# MIntRec2.0: A Large-scale Benchmark Dataset for Multimodal Intent Recognition and Out-of-scope Detection in Conversations

**Hanlei Zhang**[1*] **Xin Wang**[1,2,3*] **Hua Xu**[1†] **Qianrui Zhou**[1] **Kai Gao**[2†]
**Jianhua Su**[1,2,3] **Jinyue Zhao**[1,2] **Wenrui Li**[1] **Yanting Chen**[1]
Tsinghua University[1]  Hebei University of Science and Technology[2]
Samton (Jiangxi) Technology Development Co.,Ltd, Nanchang 330036 ,China[3]

## ABSTRACT

Multimodal intent recognition poses significant challenges, requiring the incorporation of non-verbal modalities from real-world contexts to enhance the comprehension of human intentions. However, most existing multimodal intent benchmark datasets are limited in scale and suffer from difficulties in handling out-of-scope samples that arise in multi-turn conversational interactions. In this paper, we introduce MIntRec2.0, a large-scale benchmark dataset for multimodal intent recognition in multi-party conversations. It contains 1,245 high-quality dialogues with 15,040 samples, each annotated within a new intent taxonomy of 30 fine-grained classes, across text, video, and audio modalities. In addition to more than 9,300 in-scope samples, it also includes over 5,700 out-of-scope samples appearing in multi-turn contexts, which naturally occur in real-world open scenarios, enhancing its practical applicability. Furthermore, we provide comprehensive information on the speakers in each utterance, enriching its utility for multi-party conversational research. We establish a general framework supporting the organization of single-turn and multi-turn dialogue data, modality feature extraction, multimodal fusion, as well as in-scope classification and out-of-scope detection. Evaluation benchmarks are built using classic multimodal fusion methods, ChatGPT, and human evaluators. While existing methods incorporating nonverbal information yield improvements, effectively leveraging context information and detecting out-of-scope samples remains a substantial challenge. Notably, powerful large language models exhibit a significant performance gap compared to humans, highlighting the limitations of machine learning methods in the advanced cognitive intent understanding task. We believe that MIntRec2.0 will serve as a valuable resource, providing a pioneering foundation for research in human-machine conversational interactions, and significantly facilitating related applications. The full dataset and codes are available for use at https://github.com/thuiar/MIntRec2.0.

## 1 INTRODUCTION

Understanding human intentions in multimodal scenarios holds significant research importance and has broad applications, such as human-computer interaction (Xu, 2019), intelligent transportation system (Kaffash et al., 2021), and medical diagnosis (Tiwari et al., 2022; Moon et al., 2022). For instance, perceiving user tones, expressions, and body language enables better capture of user needs in intelligent customer systems. This also leads to more personalized, efficient, and natural interactions (Luo et al., 2022). While there emerge numerous multimodal language datasets in recent years, particularly in multimodal sentiment analysis and emotion recognition (Li et al., 2019; Chudasama et al., 2022; Hu et al., 2022b), few datasets provide high-quality multimodal intent resources, which significantly hampers related research. Zhang et al. (2022a) pioneered this area by formulating intent taxonomies in multimodal conversational scenarios and providing 2,224 annotated utterances with

---

*Equal contribution.   †Corresponding authors.

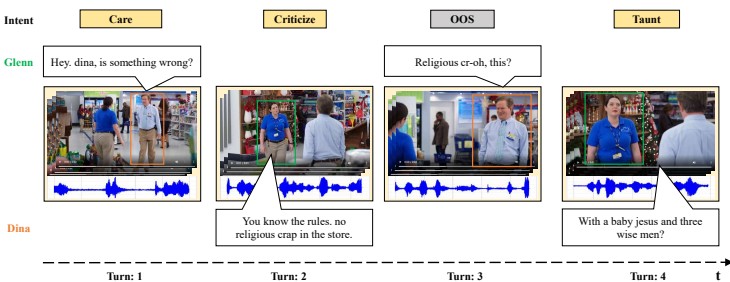

Figure 1: An example from the MIntRec2.0 dataset. More examples are provided in the Appendix A.

text, video, and audio information. However, it has three major limitations: First, its scale is relatively small compared to other multimodal datasets (Zadeh et al., 2018b; Poria et al., 2019), leading to potential overfitting and impacting the generalization ability. Second, it only includes utterances from single-turn dialogues, neglecting context and multi-party information. Third, it fails to consider out-of-scope utterances, which commonly occur in dialogue systems (Larson et al., 2019) and are crucial for improving system robustness.

To address these issues, we propose MIntRec2.0, a large-scale multimodal multi-party benchmark dataset that comprises 1,245 high-quality dialogues, totaling 12.3 hours. A representative sample is depicted in Figure 1. The construction of this dataset involves four main steps. Initially, raw videos from three TV series are collected and segmented into utterance-level portions based on timestamps. These segments are then manually grouped into dialogues in alignment with the conversational scenes and events. Subsequently, each utterance is annotated with speaker identity information to leverage specific contextual information. Following this, we propose a new intent taxonomy incorporating 30 fine-grained intent classes. An *OOS* tag is also added to identify utterances that do not belong to any known classes, a phenomenon commonly occurred in real-world, open-ended scenarios. Lastly, six experienced workers annotate each piece of data using text, video, and audio information. The final dataset contains 9,304 in-scope and 5,736 out-of-scope samples.

We develop a general framework for multimodal intent recognition and out-of-scope detection within single-turn and multi-turn conversations. First, data inputs are organized at both utterance and dialogue levels, where the latter retrieves all the context information corresponding to the speaker in the current dialogue turn. Secondly, we extract text, video, and audio features for each utterance. For multi-turn dialogues, context information is concatenated to the utterance in the current turn using a special token as a separator. Third, we perform multimodal fusion on the extracted features. Specifically, we employ two strong multimodal fusion methods (Tsai et al., 2019; Rahman et al., 2020) to leverage nonverbal information by capturing cross-modal interactions. In the training stage, in addition to the multimodal fusion loss, cross-entropy loss is applied under the supervision of hard and soft targets for learning in-scope and out-of-scope data, respectively. During inference, a threshold-based method (Shu et al., 2017) is adopted to both identify high-confidence in-scope and detect low-confidence out-of-scope samples. Experimental results demonstrate that using multimodal information can effectively improve in-scope intent recognition accuracy and enhance out-of-scope detection robustness. Furthermore, we evaluate ChatGPT and human performance under a challenging setting with few-shot samples as prior knowledge. The results reveal a significant performance gap of over 30% absolute scores between large language models (LLMs) and humans. Humans achieve the state-of-the-art benchmark performance of 71% accuracy with merely 7% of the training data, indicating this dataset is extremely challenging for existing machine learning methods.

**Contributions**. (1) This paper presents MIntRec2.0, the first large-scale multimodal multi-party conversational intent dataset. This dataset provides detailed annotations for both intent and speaker identity for each utterance within multimodal contexts and enables out-of-scope detection in open-world scenarios. (2) We establish a universal framework for in-scope classification and out-of-scope detection, applicable to both single-turn and multi-turn conversations, and introduce strong benchmark baselines. (3) Extensive experiments demonstrate the effectiveness of leveraging multimodal information in intent recognition. However, considerable opportunities for enhancement persist in existing methods when compared with human performance, highlighting the challenges inherent in high-level cognitive intent recognition tasks and underscoring the value of this dataset in advancing

Table 1: Comparison of the MIntRec2.0 dataset with previous intent datasets. #I and #U represent the number of intent classes and utterances. Conv. denotes the conversational nature of the dataset. OOS and Multi-Party indicate the inclusion of out-of-scope examples and multiple speakers per dialogue, respectively. T, V, and A represent text, video, and audio information.

| Datasets | #I | #U | Conv. Scenes | Conv. Type | OOS | Multi-Party | T | V | A |
|---|---|---|---|---|---|---|---|---|---|
| ATIS (Tür et al., 2010) | 17 | 6,371 | ✓ | Single-turn | ✗ | ✗ | ✓ | ✗ | ✗ |
| Snips (Coucke et al., 2018) | 7 | 14,484 | ✓ | Single-turn | ✗ | ✗ | ✓ | ✗ | ✗ |
| CLINC150 (Larson et al., 2019) | 150 | 23,700 | ✓ | Single-turn | ✓ | ✗ | ✓ | ✗ | ✗ |
| MDID (Kruk et al., 2019) | 7 | 1,299 | ✗ | - | ✗ | ✗ | ✓ | ✓ | ✗ |
| Intentonomy (Jia et al., 2021) | 28 | 14,455 | ✗ | - | ✗ | ✗ | ✗ | ✓ | ✗ |
| MIntRec (Zhang et al., 2022a) | 20 | 2,224 | ✓ | Single-turn | ✗ | ✗ | ✓ | ✓ | ✓ |
| MIntRec2.0 | 30 | 15,040 | ✓ | Multi-turn | ✓ | ✓ | ✓ | ✓ | ✓ |

related research. This dataset will be released under the CC BY-NC-SA 4.0 license, and codes will be publicly available as open source. A portion of the data are accessible in supplementary materials.

## 2 RELATED WORK

This section provides a brief overview of the existing literature in benchmark datasets, multimodal fusion methods, and multimodal multi-turn conversations. Further related works focusing on video understanding and intent analysis are detailed in Appendix B.

**Benchmark Datasets.** Intent recognition is a substantial task in natural language processing (NLP) and is supported by a numerous of benchmark datasets. These datasets can be broadly categorized into two branches. The first branch originates from task-oriented dialogues and includes datasets like ATIS (Tür et al., 2010), SNIPS (Coucke et al., 2018), CLINC150 (Larson et al., 2019), BANK-ING77 (Casanueva et al., 2020). Notably, CLINC150 incorporates out-of-scope data to test system robustness. SIMMC 2.0 (Kottur et al., 2021) is a multimodal dataset focusing on the shopping domain, but it lacks intent annotations. The second branch stems from open-ended dialogues, represented by multi-turn dialogue datasets such as DailyDialog (Li et al., 2017) and SWBD (Godfrey et al., 1992). However, these datasets primarily offer dialogue acts and may not be well-suited for applications requiring specific intent classes. In recent years, there has been a growing interest in multimodal language datasets for both single-turn (Zadeh et al., 2016; 2018b; Yu et al., 2020) and multi-turn dialogues (Busso et al., 2008; Poria et al., 2019). EMOTyDA (Saha et al., 2020) is another large-scale multimodal dataset for multi-turn dialogues, but it only includes coarse-grained dialogue acts. Some studies have also explored visual or multimodal intents using image modality (Jia et al., 2021; Kruk et al., 2019). MIntRec (Zhang et al., 2022a) stands as the first multimodal intent recognition dataset for open-ended dialogues. MIntRec2.0 significantly expands in scale from 2,224 to 15,040 utterances and is designed to handle both out-of-scope utterances and multi-turn dialogues. A comparison between MIntRec2.0 and other benchmark intent datasets is presented in Table 1.

**Multi-modal Fusion Methods.** Multimodal fusion presents prosperous development in multimodal language understanding. Early methods aim to learn cross-modal relations and single-modal properties (Fukui et al., 2016; Zadeh et al., 2017; 2018a; Hazarika et al., 2020) or efficient multimodal representations (Liu et al., 2018). MulT (Tsai et al., 2019) designs an effective crossmodal attention module to learn adaptations across different modalities. MAG-BERT (Rahman et al., 2020) integrates nonverbal information into pre-trained language models using a multimodal adaptation gate. MBT (Nagrani et al., 2021) restricts cross-modal information flow through tight fusion bottlenecks, facilitating the connection of relevant inputs in each modality. Very recently, TCL-MAP (Zhou et al., 2024) leverages prompt learning to provide high-quality supervised signals for multimodal representation learning. We also explore state-of-the-art methods in multimodal sentiment analysis (MSA), such as Self-MM (Yu et al., 2021) and MMIM (Han et al., 2021). However, these methods rely on specific sentiment properties (e.g., polarity) that are not applicable to our task.

**Multimodal Multi-turn Conversations.** Leveraging multimodal information is a hot topic in multi-turn conversations (Ghosal et al., 2019; Majumder et al., 2019; Ghosal et al., 2020a). For instance, DialogueRNN (Majumder et al., 2019) uses GRU networks to track important temporal informa-

Table 2: Expanded intent classes in the MIntRec2.0 dataset with brief interpretations.

| Intent Categories | | Interpretations |
|---|---|---|
| Express emotions or attitudes | Doubt | Convey a sense of mistrust or uncertainty regarding someone or something (e.g., questioning with an expression of disbelief). |
| | Acknowledge | Indicate comprehension or agreement (e.g., using affirming words such as alright, well). |
| | Refuse | Show unwillingness or rejection (e.g., using negative words to decline an offer or request). |
| | Warn | Alert to potential dangers or risks (e.g., signaling alarm with a serious expression and tone). |
| | Emphasize | Highlight the importance or significance of something (e.g., speaking with stress and a determined attitude). |
| Achieve goals | Ask for opinions | Request others' views or thoughts on a particular matter (e.g., asking for others' perspectives). |
| | Confirm | Validate or ascertain the truth or accuracy of something (e.g., affirming certainty without raising doubts). |
| | Explain | Provide an elaborate account or clarification (e.g., using explanatory and causal words such as because). |
| | Invite | Offer someone to participate in an activity or event (e.g., asking someone to join in activities like going out). |
| | Plan | Organize or schedule an event or action (e.g., deliberating on schedules and making commitments for the future). |

tion, including the history of speaker states and global states. MM-DFN (Hu et al., 2022a) proposes a graph-based dynamic fusion module to reduce historical redundancy while tracking the history of speaker states. Another approach is to construct multimodal fusion networks to integrate contextual information between different modalities, such as M2FNet (Chudasama et al., 2022) and MMGCN (Hu et al., 2021). However, modeling temporal contextual information with multimodal fusion representations does not yield good results (see Appendix C). Therefore, we propose a simple baseline that concatenates the context information of inputs before multimodal fusion.

## 3 THE MIntRec2.0 DATASET

**Data Sources & Dialogue Division.** First, we collect raw videos from three different TV series: Superstore, The Big Bang Theory, and Friends on YouTube and obtain subtitles from OpenSubtitles. We ensure the selected videos do not offend user privacy and do not contain malicious content (Appendix D). We split them into continuous video segments according to the timestamps in the transcripts and extract corresponding audio segments. Then, we organize them into a series of dialogues for multi-turn dialogue intent analysis. Specifically, we manually annotate the starting and ending indices of video segments for each dialogue and distinguish different dialogues based on whether they are in the same scene and episode, as suggested in (Poria et al., 2019). Besides, we establish a baseline to estimate the utterance boundary in each segmented dialogue (Appendix E).

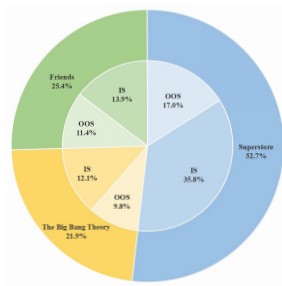

Figure 2: In-scope and out-of-scope data distribution.

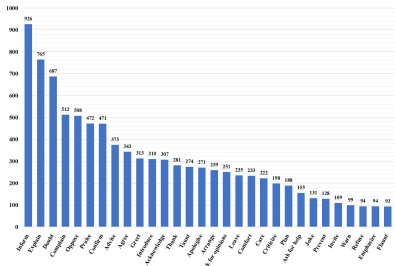

Figure 3: Distribution of in-scope intents in the MIntRec2.0 dataset.

Table 3: Data statistics. # denotes the total number.

| | |
|---|---|
| # data sources | 3 |
| # intents classes | 30 |
| # dialogues | 1,245 |
| # utterances | 15,040 |
| # in-scope utterances | 9,304 |
| # out-of-scope utterances | 5,736 |
| # words in utterances | 118,477 |
| # unique words in utterances | 9,524 |
| Average length of utterances | 7.9 |
| Maximum length of utterances | 46 |
| Average video clip duration | 3.0 (s) |
| Maximum video clip duration | 19.9 (s) |
| Video hours | 12.3 (h) |

**Speaker Information.** In multi-turn conversations, we can leverage context information to help analyze the intent conveyed by the speaker in each dialogue turn. However, context information may involve multiple speakers (e.g., there are a total of 51.5% dialogues with more than two speakers). As using context information of speakers is helpful for intent analysis (Ghosal et al., 2020b), we aim to differentiate different speakers in each dialogue and annotate the identities of the speakers. Specifically, we perform annotation of 21, 7, and 6 main characters in Superstore, The Big Bang Theory, and Friends, respectively, which account for 90.4% of the data. The remaining data include other characters with fewer appearances (Refer to Appendix F for statistics of different characters).

**Expanded Intent Classes.** In this work, we utilize the established intent taxonomy from the MIntRec dataset (Zhang et al., 2022a). However, as the dataset primarily focuses on discrete single-turn conversations, and the existing 20 intent classes are insufficient for capturing the diverse range

of intents in continuous multi-turn conversations. To address this issue, we conduct a comprehensive analysis of the divided dialogues and collect 10 additional high-frequency intent tags for the two coarse-grained intent classes (i.e., *express emotions or attitudes* and *achieve goals*). Specifically, we add *doubt, acknowledge, refuse, warn, emphasize* to the former category, and *ask for opinions, confirm, explain, invite, plan* to the latter. Interpretations of both the expanded and existing intent categories can be found in Table 2 and Appendix G, respectively. Notably, these newly introduced classes account for 37.3% of the utterances in our dataset, highlighting their significance in intent understanding. The intent taxonomies are highly applicable across various domains, offering considerable promise for real-world applications (Further discussions can be found in Appendix H).

**Out-of-scope Utterances.** As intents usually reside within particular contextual events (Schröder et al., 2014), there inevitably exist some utterances that fall outside the predefined intent categories in continuous conversational interactions, as suggested in (Larson et al., 2019). There are two common types of such utterances. First, there are statements that primarily convey factual information, such as *statement-non-opinion* defined in the 42 dialogue acts (Godfrey et al., 1992). While this type of dialogue act covers a significant proportion of utterances in multi-turn conversations, it provides limited contribution to understanding specific and applicable intents. Second, due to the diverse and uncertain nature of human intentions, the predefined intent classes cannot cover all possible intentions in an open-world environment (Zhang et al., 2023b), and there may exist utterances falling under open intent classes. Given the ambiguous boundary in determining specific out-of-scope utterances, we adopt a similar manner as in (Larson et al., 2019) and define them as those that do not belong to any of the existing intent classes. Taking these utterances into account in multi-turn conversations brings us closer to real-world scenarios and presents many practical applications.

**Annotation Process.** Six college students proficient in English are employed to perform multimodal label annotation. They are provided with a comprehensive guidebook detailing intent interpretations and application scenarios and are only permitted to begin after achieving high accuracy on seed examples. The annotators are evenly divided into two groups and assigned to annotate half of the data simultaneously. To facilitate their work, a user-friendly annotation platform with a unified database has been developed (Appendix I). Each worker is tasked with analyzing the speaker's intention in a video segment by combining text, video, and audio information. They are then required to choose from a set of 30 known intent tags, as well as an *OOS* tag. The final label for each utterance is determined through majority voting, with at least two out of three votes required to reach a consensus. We operate under the assumption that each utterance has a single intent, and the rationale for not opting for multi-intent labeling is elaborated in Appendix J. To mitigate potential issues, utterances that receive three different votes are excluded from our dataset.

**Annotation Results.** We have successfully collected 1,245 high-quality dialogues to create the MIntRec2.0 dataset. This dataset consists of 9,304 in-scope and 5,736 out-of-scope utterances with multimodal labels. The statistics of the dataset are presented in Table 3. To assess annotation reliability, we calculate the Fleiss's kappa statistics for each of our six annotators to measure interrater reliability. The Fleiss's kappa scores range from 0.66 to 0.70, averaging 0.69. This indicates a level of *substantial* agreement, as defined in (McHugh, 2012). The distribution of the dataset across three different data sources is illustrated in Figure 2. Superstore, The Big Bang Theory, and Friends contribute 53%, 22%, and 25% of the dataset, respectively. Each data source contains between 54.5% and 67.9% of in-scope utterances. The intent distribution of in-scope utterances is depicted in Figure 3, demonstrating a common long-tailed distribution similar to real-world scenarios. As expected, some intents such as *inform, explain, doubt*, and *complain* are more prevalent in daily life, while others like *warn, refuse, emphasize*, and *flaunt* tend to occur less in specific occasions and scenes. To ensure adequate training, each intent class contains more than 90 samples.

## 4 BENCHMARK FRAMEWORK

This section presents a general benchmark framework, illustrated in Figure 4. It includes data organization, multimodal feature extraction, multimodal fusion, training, and evaluation.

**Data Organization.** In the case of single-turn dialogues, we utilize the pre-segmented utterance-level samples. Each individual utterance represents a complete turn of dialogue and includes corre-

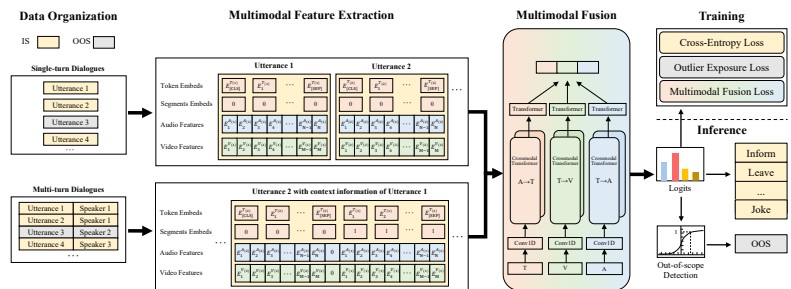

Figure 4: Overview of the benchmark framework for the MIntRec2.0 dataset.

sponding text, video, and audio information of one speaker. For multi-turn dialogues, we employ well-divided dialogues as described in Section 3. In particular, the utterances within each dialogue are arranged chronologically based on the order in which the speakers take their turn. To further leverage the context of the respective speaker, we attribute the corresponding speaker identity information to each utterance, as suggested in (Poria et al., 2019).

**Text Feature Extraction**. We select the pre-trained BERT (Devlin et al., 2018) language model as a powerful backbone for processing the text modality, which has demonstrated strong performance when fine-tuned on our dataset. For each text utterance $s$, we first tokenize it in the required format, i.e., [CLS], $s_1, \cdots, s_n$, [SEP], and then obtain the token embeddings $\mathbf{E}^T \in \mathbb{R}^{L^T \times D^T}$, where $L^T$ is the sequence length, and $D^T$ is the feature dimension.

**Video Feature Extraction**. Video features are extracted at the frame-level, as suggested in (Yu et al., 2020; Zadeh et al., 2018b). Since video frames often contain multiple individuals, we begin by identifying regions of interest (RoIs) for the speakers, using a sequence of automated procedures. This involves scene detection, object detection (Ren et al., 2015), face detection (Zhang et al., 2017), face tracking, and audio-visual active speaker detection (Tao et al., 2021), as described in (Zhang et al., 2022a). This process can generate more than 1,000K high-quality keyframes with speaker bounding boxes in approximately 5 days. Next, we use these annotated RoIs and employ the instance segmentation method, Mask R-CNN (He et al., 2017), pre-trained on the COCO (Lin et al., 2014) dataset to extract visual features. We utilize the well-initialized Swin Transformer (Liu et al., 2021), pre-trained on the ImageNet-1K (Deng et al., 2009) dataset, as the backbone due to its superior vision task performance. We use it to extract feature maps of each keyframe and apply RoIAlign (He et al., 2017) to convert them into fixed sizes using annotated RoIs. Finally, applying average pooling to these feature maps yields the overall RoI feature embeddings $\mathbf{E}^V \in \mathbb{R}^{L^V \times D^V}$.

**Audio Feature Extraction**. To process the audio modality, we first use the librosa toolkit (McFee et al., 2015) to load the audio waveform data with a sampling rate of 16,000 Hz. Then, we employ WavLM (Chen et al., 2022), a speech pre-trained model to extract audio feature representations. Due to its masked speech prediction and denoising pre-training strategy, it has shown remarkable performance in a wide range of speech tasks, outperforming other powerful speech pre-trained models such as wav2vec 2.0 (Baevski et al., 2020) and HuBERT (Hsu et al., 2021). Notably, it excels in speaker verification and speech separation tasks, which is suitable for conversational scenarios involving multiple speakers. By utilizing WavLM, we acquire audio embeddings $\mathbf{E}^A \in \mathbb{R}^{L^A \times D^A}$.

**Incorporating Context Information**. In single-turn dialogues, we can directly extract embeddings for text, video, and audio modalities, as mentioned previously. However, in multi-turn dialogues, it is substantial to consider the context information of different modalities to gain a better understanding of the conversation. To address this, we utilize the context information based on different speakers, as suggested in (Majumder et al., 2019; Ghosal et al., 2019). Specifically, for the utterance in the current turn, we first obtain the speaker identity information and then retrieval all the content from the previous dialogue turns corresponding to this speaker, which serves as the context information. Next, we employ a simple and effective method to leverage the context information by concatenating it with the utterance in the current turn. Taking the context information from one turn of utterance as an example, for the text modality, the first sequence comprises all the token embeddings in the

current turn: $\mathbf{E}_{[\text{CLS}]}^{T_{(1)}}, \mathbf{E}_1^{T_{(1)}}, \cdots, \mathbf{E}_2^{T_{(1)}}, \mathbf{E}_{[\text{SEP}]}^{T_{(1)}}$. The second sequence comprises the context information. We remove the first token [CLS] and concatenate the remaining embeddings with the first sequence: $\mathbf{E}_{[\text{CLS}]}^{T_{(1)}}, \cdots, \mathbf{E}_{[\text{SEP}]}^{T_{(1)}}, \mathbf{E}_1^{T_{(2)}}, \cdots, \mathbf{E}_{[\text{SEP}]}^{T_{(2)}}$. Besides, we include segment embeddings to aid in understanding the relationships between current and contextual utterances. The segment embeddings for the first and second sequences are encoded as zero and one vectors, respectively, with the same length as the token embeddings. For nonverbal modalities, we insert a one-dimensional zero vector between the feature embeddings of the two sequences to distinguish them. If additional context information is available, such as more contextual utterances, we append each of them to the end of the latest context utterance using the same operation as the second sequence.

**Multimodal Fusion**. After extracting multimodal features, our goal is to utilize multimodal fusion techniques to capture cross-modal interactions and exploit complementary information from different modalities to further enhance intent recognition capability. Specifically, we use $\mathbf{E}^T$, $\mathbf{E}^V$, and $\mathbf{E}^A$ as inputs and feed them into a multimodal fusion network $\mathcal{F}$ to obtain multimodal representations $\mathbf{z} = \mathcal{F}(\mathbf{E}^T, \mathbf{E}^V, \mathbf{E}^A)$. In this work, we adopt two strong multimodal fusion methods, namely MAG-BERT (Rahman et al., 2020) and MulT (Tsai et al., 2019) as baselines.

**Training**. Following multimodal fusion, we employ the multimodal representations $\mathbf{z}$ for training. For in-scope samples $\mathbf{z}^{\text{in}} = \{\mathbf{z}_i | y_i \in \mathcal{Y}\}_{i=1}^N$, we perform classification on $\mathbf{z}^{\text{in}}$ using the cross entropy loss $\mathcal{L}_{\text{CE}}$, where $N$ is the number of training samples, and $\mathcal{Y}$ is the set of $K$ known intent labels. For out-of-scope samples $\mathbf{z}^{\text{out}} = \{\mathbf{z}_i | y_i \notin \mathcal{Y}\}_{i=1}^N$, we apply the outlier exposure (OE) (Hendrycks et al., 2018) loss, denoted as $\mathcal{L}_{\text{OE}}$, to distinguish them from the in-scope samples and enhance the model's robustness and its generalization ability for out-of-scope samples. Specifically, we use a uniform distribution over the $K$ known classes as soft targets. The definitions for losses are as follows:

$$\mathcal{L}_{\text{CE}} = -\frac{1}{N}\sum_{i=1}^N \log \frac{\exp(\phi(\mathbf{z}_i^{\text{in}})^{y_i})}{\sum_{j=1}^K \exp(\phi(\mathbf{z}_i^{\text{in}})^j)}, \mathcal{L}_{\text{OE}} = -\frac{1}{N}\sum_{i=1}^N\sum_{j=1}^K \frac{1}{K} \log \frac{\exp(\phi(\mathbf{z}_i^{\text{out}})^j)}{\sum_{m=1}^K \exp(\phi(\mathbf{z}_i^{\text{out}})^m)},$$

where $\phi(\cdot)$ is the classifier with a linear layer. The training loss $\mathcal{L}_{\text{Train}} = \mathcal{L}_{\text{CE}} + \mathcal{L}_{\text{OE}} + \mathcal{L}_{\text{Fusion}}$, where $\mathcal{L}_{\text{Fusion}}$ is the loss specified in different multimodal fusion methods. Besides, we also conduct experiments by training a $(K+1)$-way classifier with out-of-scope samples grouped as the $(K+1)^{\text{th}}$ class, resulting in significant decrease in the performance of in-scope classification (Appendix K).

**Inference**. During inference, our goal is to both identify in-scope classes and detect out-of-scope samples. To accomplish this, we employ a threshold-based open world classification method in NLP called DOC (Shu et al., 2017), which performs well in our experiments. This method rejects low-confidence samples, assigning statistical thresholds to each known class. For each sample $\mathbf{z}_i$, the predicted probability of the $k^{\text{th}}$ class is given by $p(k|\mathbf{z}_i) = \text{Sigmoid}(\phi(\mathbf{z}_i)^k)$. We use the output probabilities from each class of the training samples to calculate the corresponding class threshold $\delta_k$. Specifically, we fit them to one half of the Gaussian distribution with $\mu = 1$ and calculate the standard deviations $\sigma_k$ using two symmetric halves of the probabilities. The class threshold is then given by $\delta_k = \max(0.5, 1 - \alpha\sigma_k)$, where $\alpha = 1$ usually works well. A test sample is detected as out-of-scope if $p(k|\mathbf{z}_i) < \delta_k, \forall k \in \mathcal{Y}$. Otherwise, it is considered as an in-scope sample and is assigned the predicted class with the maximum probability, denoted as $y_p = \text{argmax}_{k \in \mathcal{Y}} p(k|\mathbf{z}_i)$.

## 5 EXPERIMENTS

**Implementation Details**. We partition our dataset into training, validation, and testing sets, maintaining an approximate ratio of 7:1:2 for both dialogues and utterances. (Further details are provided in Appendix L). For the text modality, we utilize BERT$_{\text{LARGE}}$ as a powerful backbone consisting of 24 transformer layers implemented in the Huggingface transformers library (Wolf et al., 2020), to extract features with the dimension $D^T$ of 1024. For the video modality, we employ well-trained checkpoints of Mask R-CNN from the MMDetection toolbox (Chen et al., 2019) to extract features with the dimension $D^V$ of 256. For the audio modality, we use the pre-trained model WavLM, implemented in (Wolf et al., 2020) to extract features with the dimension $D^A$ of 768. In single-turn dia-

Table 4: Benchmark baseline results on the MIntRec2.0 dataset.

| Train | Methods | In-scope Classification | | | | | | In-scope + Out-of-scope Classification | | | |
|---|---|---|---|---|---|---|---|---|---|---|---|
| | | F1 | P | R | ACC | WF1 | WP | F1-IS | ACC | F1-OOS | F1 |
| w / o OOS | TEXT | 51.60 | 55.47 | 51.31 | 59.30 | 58.01 | 58.85 | 43.37 | 43.24 | 30.40 | 42.96 |
| | MAG-BERT | 55.17 | 57.78 | 55.10 | 60.58 | 59.68 | 59.98 | 46.48 | 44.80 | 34.03 | 46.08 |
| | Δ(MAG-BERT) | 3.57↑ | 2.31↑ | 3.79↑ | 1.28↑ | 1.67↑ | 1.13↑ | 3.11↑ | 1.56↑ | 3.63↑ | 3.12↑ |
| | MulT | 54.12 | 58.02 | 53.77 | 60.66 | 59.55 | 60.12 | 45.65 | 46.14 | 38.57 | 45.42 |
| | Δ(MulT) | 2.52↑ | 2.55↑ | 2.46↑ | 1.36↑ | 1.54↑ | 1.27↑ | 2.28↑ | 2.90↑ | 8.17↑ | 2.46↑ |
| w OOS | TEXT | 52.08 | 54.57 | 52.11 | 59.99 | 58.62 | 58.65 | 45.83 | 55.61 | 61.54 | 46.34 |
| | MAG-BERT | 53.64 | 54.84 | 53.79 | 60.12 | 59.11 | 58.83 | 47.52 | 56.20 | 62.47 | 48.00 |
| | Δ(MAG-BERT) | 1.56↑ | 0.27↑ | 1.68↑ | 0.13↑ | 0.49↑ | 0.18↑ | 1.69↑ | 0.59↑ | 0.93↑ | 1.66↑ |
| | MulT | 52.72 | 56.45 | 52.56 | 60.18 | 58.82 | 59.38 | 46.88 | 56.00 | 61.66 | 47.35 |
| | Δ(MulT) | 0.64↑ | 1.88↑ | 0.45↑ | 0.19↑ | 0.20↑ | 0.73↑ | 1.05↑ | 0.39↑ | 0.12↑ | 1.01↑ |
| w OOS | Context TEXT | 53.61 | 54.46 | 54.10 | 59.04 | 58.69 | 59.27 | 46.42 | 56.12 | 63.56 | 46.98 |
| | Context MAG-BERT | 53.89 | 55.72 | 54.21 | 59.84 | 59.41 | 60.22 | 46.74 | 56.20 | 62.52 | 47.25 |
| | Δ(Context MAG-BERT) | 0.28↑ | 1.26↑ | 0.11↑ | 0.80↑ | 0.72↑ | 0.95↑ | 0.32↑ | 0.08↑ | 1.04↓ | 0.27↑ |
| | Context MulT | 53.96 | 54.91 | 54.15 | 59.48 | 59.33 | 60.04 | 46.45 | 56.07 | 62.93 | 46.98 |
| | Δ(Context MulT) | 0.35↑ | 0.45↑ | 0.05↑ | 0.44↑ | 0.64↑ | 0.77↑ | 0.03↑ | 0.05↓ | 0.63↓ | 0.00 |

logues, we apply zero-padding with a maximum sequence length of 50, 180, and 400 for text, video, and audio features, respectively. The number of training epochs is set to 40, and the training batch size is set to 16 for all baselines. We employ AdamW (Loshchilov & Hutter, 2019) for optimization, implement our approach using PyTorch 1.13.1, and conduct experiments on Tesla V100-SXM2-32GB GPUs. For all experiments, we report the results averaged over five runs, using random seeds ranging from 0 to 4 (Additional hyper-parameters details are available in Appendix M).

**Benchmark Baselines**. As text is the predominant modality in conversational multimodal intent recognition (Zhang et al., 2022a), we establish a robust baseline by fine-tuning BERT$_{LARGE}$, comparing its performance with two multimodal fusion methods: MAG-BERT and MulT. We evaluate these methods in both single-turn and multi-turn conversations, focusing on in-scope classification and out-of-scope detection. For single-turn conversations, we use only in-scope utterances for training. The out-of-scope utterances are included in the testing set and treated as a separate class, following (Lin & Xu, 2019; Zhang et al., 2023b). For multi-turn conversations, we consider both in-scope and out-of-scope samples at the dialogue-level during training, and all the baselines utilize the context information as described in section 4. We conduct additional baselines related to dialogue intent classification in NLP and out-of-distribution detection across different sources in Appendices N and O, respectively. Besides, we test the performance of ChatGPT on our dataset using both zero-shot and few-shot settings. In the zero-shot setting, ChatGPT is provided with the prompts of the label sets (e.g., 30 intent labels and one *OOS*) and an introduction to the task. In the few-shot setting, we use 10 dialogues with 227 utterances that cover all intent classes as the learning data (Details of the utilized prompts can be found in Appendix P). Finally, we invite ten evaluators to assess human performance. Each worker is assigned an equal portion of the testing set, ensuring they have not seen the data before. They receive the same background information as that provided to ChatGPT to ensure a fair comparison. Additionally, we provide them with more prior knowledge, consisting of 100 dialogues and 997 utterances, to explore human potential in addressing this complex problem. We average the predictions from all evaluators to obtain the final score.

**Evaluation Metrics**. To evaluate the in-scope classification performance, we adopt six commonly used metrics: F1-score (F1), Precision (P), Recall (R), Accuracy (ACC), Weighted F1 (WF1), and Weighted Precision (WP). To evaluate out-of-scope detection performance, we utilize four metrics commonly employed in open intent classification (Shu et al., 2017; Zhang et al., 2023b): Accuracy, Macro F1-score over all classes, In-scope classes (F1-IS), and the Out-of-scope class (F1-OOS).

**Results**. The performance of benchmark baselines on the MIntRec2.0 dataset is presented in Table 4. In this table, Δ denotes the improvements achieved by multimodal fusion methods compared to the text baseline using the current evaluation metric. For single-turn dialogues, we conduct experiments on two settings: training without out-of-scope samples (w / o OOS) and with out-of-scope samples (w OOS). It is evident that when only in-scope utterances are available, all multimodal fusion methods significantly outperform the text baseline. MAG-BERT and MulT demonstrate 1~4% increase in scores across all metrics. Moreover, we ob-

serve that multimodal fusion methods show a larger proportion of improvement of over 2% increase in almost all settings when involving out-of-scope samples. This suggests that modeling cross-modal interactions and utilizing complementary information not only enhances in-scope identification but also remarkably improves the model robustness of out-of-scope detection.

After using out-of-scope data for training, we find that all baselines may suffer a slight decrease in some in-scope evaluation metrics but gain significant improvements in out-of-scope detection with an increase of over 30% in F1-OOS scores. Though incorporating multimodal information brings improvements on all metrics, they show less increase compared with the former setting, indicating the challenges of effectively utilizing multimodal information on out-of-scope data. For multi-turn dialogues, multimodal fusion methods yield improvements in all metrics for in-scope classification

Table 5: Performance of ChatGPT and humans on the MIntRec2.0 dataset.

| Methods | In-scope | | | In-scope + Out-of-scope | | |
|---|---|---|---|---|---|---|
| | ACC | WF1 | WP | ACC | F1-OOS | F1 |
| MAG-BERT-10 | 9.82 | 11.58 | 13.34 | 34.28 | 50.57 | 3.75 |
| ChatGPT-0 | 35.27 | 37.10 | 48.22 | 27.68 | 21.21 | 28.34 |
| ChatGPT-10 | 34.53 | 36.39 | 49.27 | 29.72 | 27.85 | 28.41 |
| Humans-10 | 64.34 | 67.82 | 72.80 | 60.43 | 62.83 | 57.83 |
| Humans-100 | **71.03** | **75.63** | **81.83** | **71.86** | **75.41** | **69.49** |

compared with the text baseline. However, it shows minor improvements or even decrease when testing on a mixture of in-scope and out-of-scope data. This also indicates that there remain substantial opportunities to explore the potential of multimodal information in conversational contexts.

**ChatGPT v.s. Humans**. Finally, we present the performance of ChatGPT and humans in Table 5. As humans typically excel at learning from few-shot samples and quickly grasping new concepts (Lake et al., 2015), we apply a challenging setting with only 10 dialogues of 227 utterances. Multimodal fusion baselines, such as MAG-BERT-10, struggle significantly in this setting, easily overfitting and falling into trivial solutions, such as predicting the most frequent in-scope or out-of-scope class, due to the challenges posed by imbalanced and few-shot training samples. In contrast, ChatGPT demonstrates much better performance even without prior knowledge of labeled data (ChatGPT-0), which shows its strong language understanding and reasoning capabilities, comprehending complex textual semantics and understanding human intentions (Bang et al., 2023). Besides, ChatGPT shows overall improvements with a 1∼6% score increase across most metrics with only 10 dialogues for training (ChatGPT-10). This suggests that ChatGPT can learn from prior knowledge and enhance intent recognition capability. Notably, it achieves a significant 6% improvement in F1-OOS, highlighting its improved out-of-scope detection robustness. However, when humans are provided with the same prior knowledge of 10 dialogues (Humans-10), they achieve an increase of over 30% in scores across almost all metrics compared to ChatGPT. This demonstrates that humans can effectively leverage limited multimodal information to understand high-level intentions and discern between known and unknown boundaries, highlighting the significant limitations of existing AI methods in this challenging task. To further explore human potential, we observe their performance with additional knowledge of 100 dialogues of 997 utterances (Humans-100). Compared with Humans-10, they achieve over a 10% improvement in almost all metrics and achieve the state-of-the-art benchmark performance. This also underscores the advantages of humans in mastering this complex task by leveraging multimodal information.

## 6 CONCLUSIONS

This paper presents MIntRec2.0, a pioneering dataset for multimodal intent recognition, encompassing 1,245 dialogues and 15,040 multimodal utterances. This marks MIntRec2.0 as the first large-scale dataset in this domain. The dataset includes annotations for speaker identity and introduces a comprehensive taxonomy of 30 intent classes, spanning 9,304 in-scope utterances. To evaluate model robustness, 5,736 out-of-scope utterances are also annotated. We propose a general framework for organizing data, extracting multimodal features, and performing multimodal fusion for in-scope classification and out-of-scope detection in both single-turn and multi-turn conversations. Extensive experiments reveal the substantial potential of using multimodal information and uncover significant opportunities for improvement in effectively utilizing out-of-scope data and context information. Moreover, even with a strong LLM such as ChatGPT, using text-only modality remains challenging in scenarios with limited prior knowledge, highlighting the importance and challenge of using multimodal information compared to human performance. The limitations and potential negative societal impacts of this work are discussed in Appendix Q.

ACKNOWLEDGMENTS

This work was supported by the National Natural Science Foundation of China (Grant No. 62173195), the National Science and Technology Major Project towards the new generation of broadband wireless mobile communication networks of Jiangxi Province (03 and 5G Major Project of Jiangxi Province) (Grant No. 20232ABC03402), High-level Scientific and Technological Innovation Talents "Double Hundred Plan" of Nanchang City in 2022 (Grant No. Hongke Zi (2022) 321-16), and supported by Hebei Natural Science Foundation (Grant No. F2022208006). We would like to thank Jiayan Teng, Zhaochen Yang, Shaojie Zhao, and Hao Li for their efforts during dataset construction.

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

## A  SAMPLE SELECTION WITHIN THE MINTREC2.0 DATASET

Figure 5 illustrates a diverse selection of samples from our MIntRec2.0 dataset to showcase representative examples. The selected samples cover all 30 intent categories and the *OOS* label.

## B  ADDITIONAL RELATED WORK

**Video Understanding**. As a significant research field within computer vision, video understanding involves the extraction of valuable information from video content. Numerous methods have been developed to handle spatial and temporal data in videos, including the Two-Stream method, which comprises TDD (Wang et al., 2015), LRCN (Donahue et al., 2015), Fusion (Feichtenhofer et al., 2016), and TSN (Wang et al., 2016). This methodology integrates a secondary path to learn a video's temporal information by training a convolutional neural network on the optical flow stream. However, these methods require extensive computation and storage capacity due to the pre-computation of optical flow.

To address this, researchers introduce 3D convolutional neural networks (3D CNNs) such as I3D (Carreira & Zisserman, 2017), R3D (Hara et al., 2018), S3D (Xie et al., 2018), Non-local (Wang et al., 2018a), and SlowFast (Feichtenhofer et al., 2019). More recently, self-attentive mechanisms like TimeSformer (Bertasius et al., 2021) and Video Swin Transformer (Liu et al., 2022) are demonstrating exceptional performance in image and video tasks. TimeSformer encodes video frames into a sequence of two-dimensional images, employing temporal self-attention to understand temporal relationships, while Video Swin Transformer partitions the input video into two-dimensional spatial and one-dimensional temporal patches, applying self-attention and cross-attention to manage long-distance temporal dependencies. X-CLIP (Ni et al., 2022), a CLIP-based method, has achieved state-of-the-art performance in video understanding by processing video content through matching video frames with text data.

While these techniques show proficiency in action recognition, they encounter difficulties when attempting to understand fine-grained intentions with high-level semantics and require considerable computational resources. For instance, X-CLIP demonstrates subpar performance on our task and demands a substantial amount of GPU memory, underscoring the need to incorporate other modalities such as language and acoustics in multimodal intent recognition tasks. Consequently, we have established baselines using multimodal fusion methods in this work.

**Intent Analysis**. Intent analysis is an important research area in spoken language understanding (Qin et al., 2021). It plays a pivotal role in task-oriented dialogue systems, enabling the recognition of user queries' intentions alongside the slot filling task (Wang et al., 2018b; Zhang et al., 2019). However, early research usually focus on the closed-world classification problem, lacking the capability to handle out-of-scope utterances encountered in real-world scenarios (Zhang et al., 2021a). To address this challenge, Lin & Xu (2019) first explore this task by employing margin loss to detect unknown intent. Zhang et al. (2021b) learn adaptive decision boundaries for each known class, thereby further reducing the open space risk. Yan et al. (2020) use Gaussian mixture models to tackle this problem and extends the task to zero-shot intent detection. Cheng et al. (2022) construct out-of-scope samples using manifold mixup technologies and employed soft labels for representation learning. Zhou et al. (2022) enhance intent representations to balance both empirical and open space risks with the aid of contrastive learning in the K-nearest neighbors space.

In practical applications, out-of-scope utterances may contain multiple fine-grained intent classes, making the discovery of potential new intent classes highly valuable for industry applications, such as dialogue and user-modeling systems (Lin et al., 2020; Li et al., 2022). Lin et al. (2020) formulate this task in a semi-supervised manner, with limited labeled data for known intents and a vast amount of unlabeled data for both known and new intents. To address this task, Lin et al. (2020); Zhang et al. (2022b) identify group-level known and new intent clusters by learning from both strong and weak pairwise supervised signals. Zhang et al. (2021c); Zhou et al. (2023) employ centroid-based alignment strategies to generate high-quality and specific pseudo-labels for self-supervised learning. However, these methods perform poorly in purely unsupervised scenarios. However, these methods have shown limited success in purely unsupervised scenarios. Zhang et al. (2023a) propose a groundbreaking approach in unsupervised new intent discovery utilizes unsupervised pre-training with strongly augmented data, followed by effective clustering. This method leverages historical centroid information for initialization and employs cluster assignments to learn discriminative representations at both the instance and cluster levels, marking a significant advancement over previous state-of-the-art methods.

## C   PERFORMANCE OF DIALOGUERNN

Table 6: Results of DialogueRNN on the MIntRec2.0 dataset.

| Setting | In-scope Classification | | | | | | In-scope + Out-of-scope Classification | | | |
|---------|------|------|------|------|------|------|-------|-------|--------|------|
|         | F1   | P    | R    | ACC  | WF1  | WP   | F1-IS | ACC   | F1-OOS | F1   |
| $K$+1   | 0.67 | 0.58 | 3.34 | 10.7 | 2.15 | 1.77 | 0.36  | 16.65 | 34.82  | 1.47 |
| Outlier Exposure | 2.75 | 4.19 | 3.74 | 3.89 | 3.23 | 5.29 | 2.21 | 11.10 | 23.67 | 2.91 |

To leverage context information, existing methods typically use multimodal fusion representations to directly model the temporal information of contexts. However, we find this approach to be ineffective for our task. Specifically, we select DialogueRNN (Majumder et al., 2019), a method specifically designed for multimodal emotion detection in conversations, for evaluation. We conduct experiments under two settings: $K$+1 and Outlier Exposure. The former treats the out-of-scope class as the $(K+1)^{\text{th}}$ class and trains using both $K$ intent classes and one out-of-scope class, while the latter employs the outlier exposure loss on out-of-scope data during training.

As illustrated in Table 6, DialogueRNN demonstrates significantly low performance across all metrics. Furthermore, we observe that it tends to fall into trivial solutions, predominantly predicting most utterances as the out-of-scope class. This observation suggests that leveraging temporal information with fused multimodal representations remains a considerable challenge. Consequently, we adopt a simple method to leverage context information by concatenating the context information from the inputs of each modality.

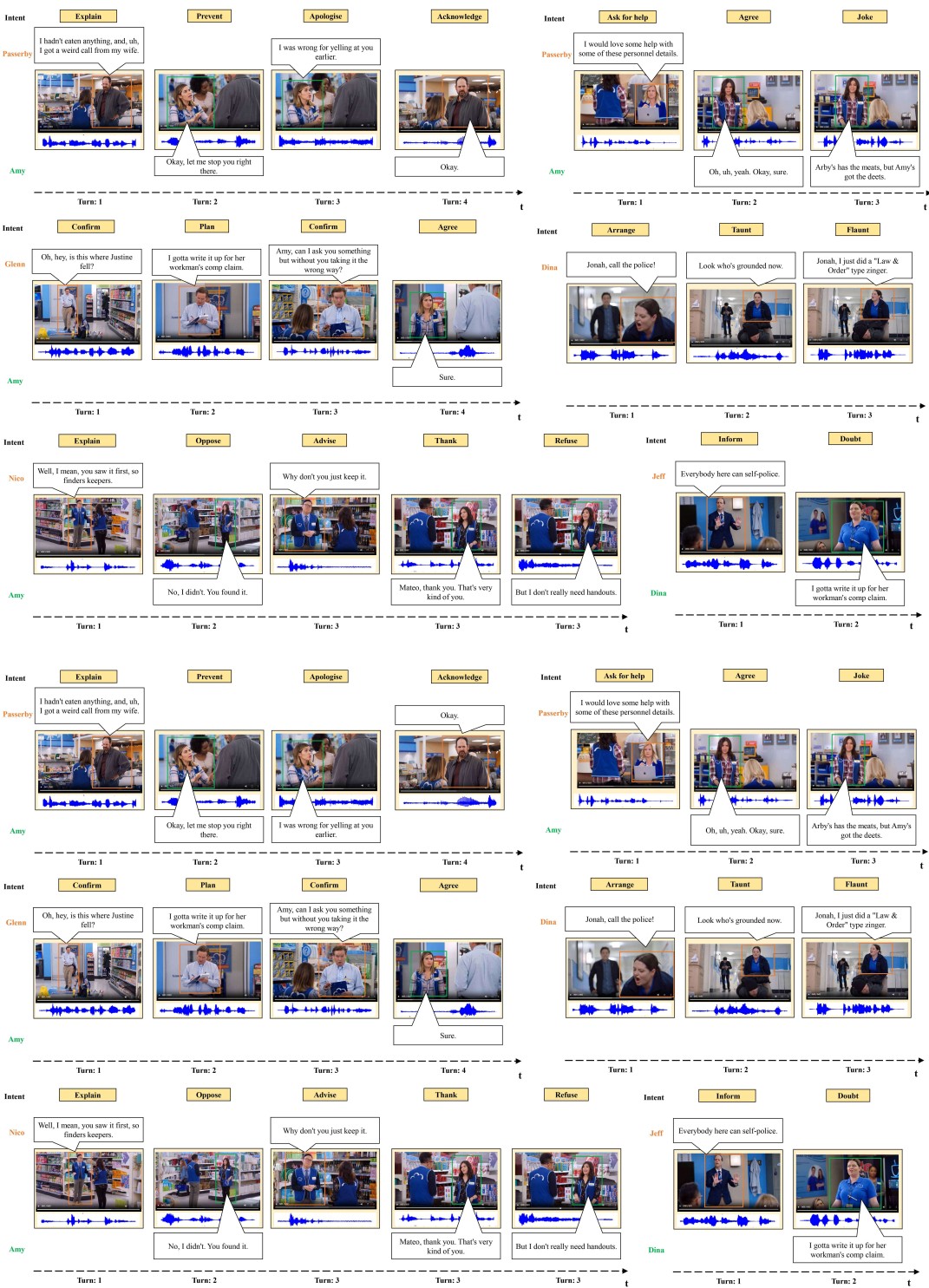

Figure 5: Samples of the MIntRec2.0 dataset.

## D  DATA PRIVACY AND CONTENT CONSIDERATIONS

Our dataset is meticulously curated and consists exclusively of character names and dialogues sourced from television shows, ensuring no infringement on the privacy or disclosure of personal information pertaining to real individuals. We have rigorously reviewed the content to maintain a high standard of decorum, assiduously avoiding any material that could be construed as offensive. Our focus remains strictly confined to the dialogues and interactions, all contextualized within the narrative framework of the respective shows, allowing for a comprehensive understanding of character dynamics without compromising ethical standards.

## E  UTTERANCE BOUNDARY ESTIMATION

To further validate the accuracy of these boundaries, we conduct additional experiments using a metric known as Speaker Boundary Error Rate (SBER), commonly employed in speech diarization tasks (Sturm et al., 2007). This metric quantifies the difference between predicted and reference speaker boundaries, with a lower SBER indicating better performance and serving as a proxy for sentence boundary accuracy. We utilize an end-to-end method implemented with pyannote (Bredin et al., 2020; Bredin & Laurent, 2021), a pre-trained speaker change detection model, to predict speaker IDs, starting times, and durations for each utterance within a dialogue segment. These predictions are then compared to the ground truth.

The results show an average SBER of 0.59 across all dialogues, suggesting considerable room for improvement in automatic sentence boundary segmentation. We believe this approach offers a reasonable method for evaluating utterance boundary performance.

## F  STATISTICS OF CHARACTERS

To further analyze the character distribution in each of the three data sources (i.e., Superstore, Friends, The Big Bang Theory) within our dataset, we present the proportions of characters from these sources in Figure 6, Figure 7, and Figure 8.

In Superstore, seven main characters and 21 recurring characters are observed. It can be noted that the seven main characters represent a significant proportion of nearly 80%, distributed uniformly. Friends have six main characters who constitute about 85% of the data, also distributed uniformly. The Big Bang Theory has seven main characters, while their distribution is imbalanced, a property we preserve due to the distinctive nature of each speaker. It is worth noting that there are other characters involved in the conversations, contributing 9.3%, 14.4%, and 5.9% respectively in each of the three TV series. These characters are also differentiated within each dialogue in our experiments.

## G  INTENT TAXONOMIES DEFINED IN THE MINTREC DATASET

The MIntRec dataset (Zhang et al., 2022a) introduces a hierarchical intent taxonomy, including two coarse-grained and 20 fine-grained intent categories. The two coarse-grained classes include *Express Emotions or Attitudes* and *Achieve Goals*. Based on these, it further includes 11 and 9 fine-grained classes for them, respectively. In particular, *Express Emotions or Attitudes* contains *complain, praise, apologize, thank, criticize, care, agree, oppose, taunt, flaunt*, and *joke*. *Achieve Goals* contains *inform, advise, arrange, introduce, comfort, leave, prevent, greet*, and *ask for help*. The interpretations of these categories are shown in Table 7, referring to (Zhang et al., 2022a).

## H  APPLICATION OF INTENT LABELS

Our intent labels can be generalized to many domains, including intelligent customer service, healthcare, mental health therapy, hazard detection, virtual assistants, and personalized recommendation systems. For instance:

- *complain, criticize, comfort*: These labels are instrumental in identifying potential mental health concerns in patients and can be pivotal in healthcare settings.

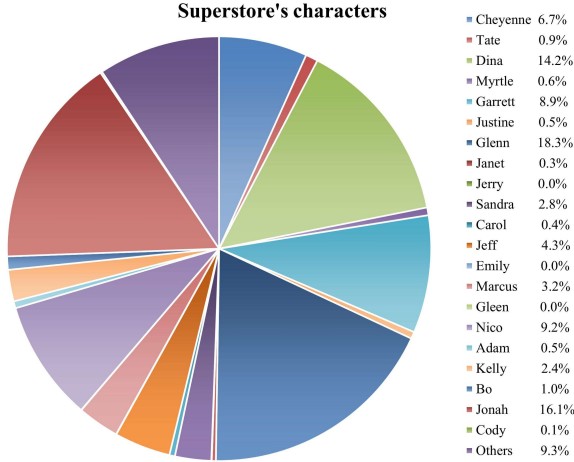

Figure 6: Proportions of characters from the TV series of Superstore.

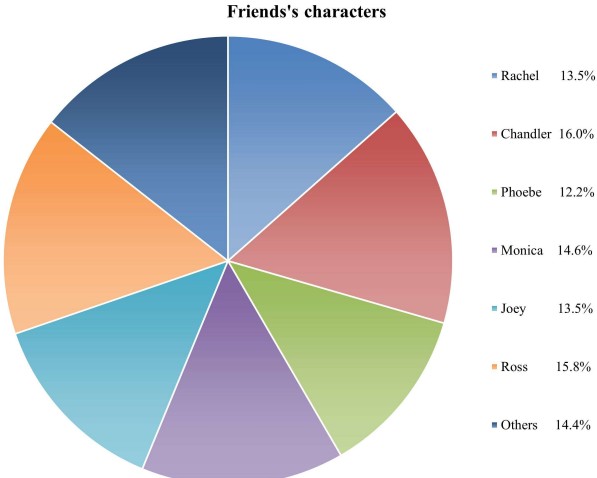

Figure 7: Proportions of characters from the TV series of Friends.

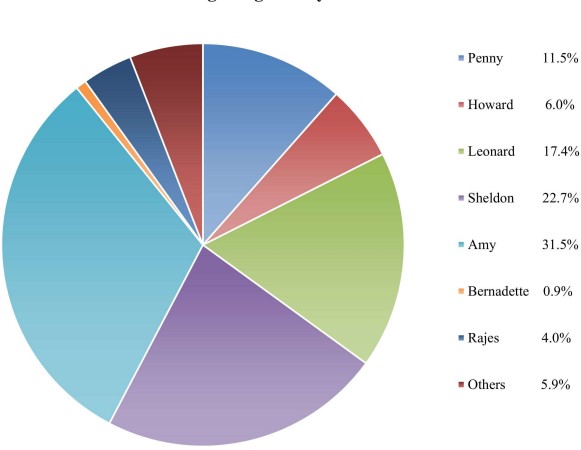

Figure 8: Proportions of characters from the TV series of The Big Bang Theory.

- *warn, prevent, OOS*: These labels can be employed effectively in systems designed for hazard detection.
- *ask for help, inform*: These labels are particularly suited for customer service platforms.
- *praise, complain, agree*: These labels can be harnessed in personalized recommendation engines.
- *the majority of these intent labels*: These labels are ideal for virtual robots designed to interact naturally with users.

Table 7: Intent taxonomies of the MIntRec dataset with brief interpretations.

| Intent Categories | | Interpretations |
|---|---|---|
| Express emotions or attitudes | Complain | Express dissatisfaction with someone or something (e.g., saying unfair encounters with a sad expression and helpless motion). |
| | Praise | Express admiration for someone or something (e.g., saying with an appreciative expression). |
| | Apologize | Express regret for doing something wrong (e.g., saying words of apology such as sorry). |
| | Thank | Express gratitude in word or deed for the convenience or kindness given or offered by others (e.g., saying words of appreciation such as thank you). |
| | Criticize | Point out and emphasize someone's mistakes (e.g., yelling out someone's problems). |
| | Care | Concern about someone or be curious about something (e.g., worrying about someone's health). |
| | Agree | Have the same attitude about something (e.g., saying affirmative words such as yeah and yes). |
| | Oppose | Have an inconsistent attitude about something (e.g., saying negative words to express disagreement) |
| | Taunt | Use metaphors and exaggerations to accuse and ridicule (e.g., complimenting someone with a negative expression). |
| | Flaunt | Boast about oneself to gain admiration, envy, or praise (e.g., saying something complimentary about oneself arrogantly). |
| | Joke | Say something to provoke laughter (e.g., saying something funny and exaggerated with a cheerful expression). |
| Achieve goals | Inform | Tell someone to make them aware of something (e.g., broadcasting something with a microphone). |
| | Advise | Offer suggestions for consideration (e.g., saying words that make suggestions). |
| | Arrange | Plan or organize something (e.g., requesting someone what they should do formally). |
| | Introduce | Communicate to make someone acquaintance with another or recommend something (e.g., describing the identify of a person or the properties of an object). |
| | Comfort | Alleviate pain with encouragement or compassion (e.g., describing something is hopeful). |
| | Leave | Get away from somewhere (e.g., saying where to go while turning around or getting up). |
| | Prevent | Make someone unable to do something (e.g., stop someone from doing something with a hand). |
| | Greet | Express mutual kindness or recognition during the encounter (e.g., waving to someone and saying hello). |
| | Ask for help | Request someone to help (e.g., asking someone to deal with the trouble). |

## I  MULTIMODAL INTENT ANNOTATION PLATFORM

We have developed an efficient platform featuring a unified database for multimodal label annotation, aiming to facilitate seamless interaction between annotators and the diverse set of multimodal data. The interface of this platform is depicted in Figure 9. This user-friendly interface allows annotators to access transcripts and associated videos from the dialogues and data sources easily, thereby ensuring accurate and consistent annotations. Annotators simply need to select one label from the 30 intent classes and an out-of-scope (OOS) tag by clicking a button. This intuitive design minimizes the learning curve for annotators and accelerates the annotation process. Once annotation is complete, the selected labels are automatically recorded in the database for statistical analysis.

This systematic approach ensures the reliability and consistency of the annotated data, which is crucial for training robust and high-performing models. The platform not only aids in the efficient collection of annotated data but also serves as a valuable tool for exploring and understanding the intricate relationships between different modalities and intents.

## J  SINGLE-INTENT ASSUMPTION

In real-world scenarios, it is possible for multiple intents coexist among the 30 pre-defined classes in a single utterance. In this work, we obey the single-intent assumption due to the following two reasons:

- **Single vs. Multi-Intent Datasets**: Most existing single-turn intent datasets in NLP, such as SNIPS, CLINC, and BANKING, focus on single-intention labeling. This is also true for multi-turn dialogue datasets like SWBD (Godfrey et al., 1992) and DailyDialog (Li et al., 2017), which generally assume a single dialogue act label at the utterance level. Therefore,

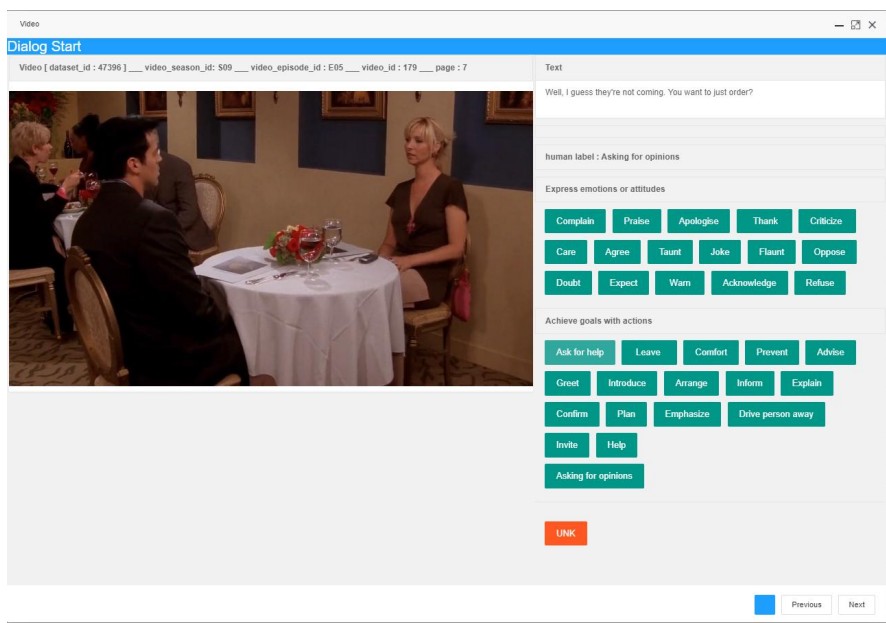

Figure 9: The interface of the annotation platform.

while multiple intentions could theoretically exist in an utterance, the prevailing practice is to identify a primary intent for the sake of clarity and brevity.

- **Applicability to Real-World Scenarios**: We have examined multi-intent datasets like Standford_LU (Hou et al., 2021) and (Xu & Sarikaya, 2013). These datasets often include action and slot labels (e.g., find music or movie, request address or route), which are more suited for task-oriented dialogue systems. Such labeling is generally not applicable in real-world multimodal scenarios, as suggested in (Zhang et al., 2022a).

To verify our assumption, we conduct an additional multi-intent annotation on the testing set. Six annotators are asked to identify up to three probable intents for each utterance. The results are shown in Table 8.

Table 8: Statistics of multiple intents in one utterance.

| Express emotions or attitudes | Classes | complain, praise, apologize, thank, criticize, care, agree, warn |
| | Number | 9, 5, 2, 1, 8, 1, 6, 1, |
| | Classes | oppose, taunt, flaunt, joke, doubt, acknowledge, refuse, emphasize |
| | Number | 7, 4, 1, 2, 14, 3, 1, 8 |
| Achieve goals | Classes | inform, advise, arrange, introduce, comfort, leave, prevent |
| | Number | 5, 1, 1, 1, 2, 4, 0 |
| | Classes | greet, ask for help, ask for opinions, confirm, explain, invite, plan |
| | Number | 1, 1, 4, 5, 35, 1, 2 |

The results show that only 136 out of 3,230 utterances (4.2%) have a second most probable intent, and none have a third. This suggests that multi-intent scenarios are relatively rare, reinforcing the adequacy of our single-intent taxonomy. In summary, our findings align with those of most existing benchmark intent datasets, indicating that our intent taxonomy is both general and distinguishable enough for real-world applications.

## K    $(K+1)$-WAY CLASSIFICATION PERFORMANCE

We also investigate another prevalent method, the $(K+1)$-way classification, to utilize the out-of-scope samples during training. In other words, we train on both the $K$ known classes and one

out-of-scope class. The results of this approach are displayed in Table 9. A noticeable decrease of approximately 10% in in-scope classification performance across numerous metrics (e.g., F1-score, recall, accuracy, weighted F1) is observed, compared to the results obtained with outlier exposure (OE) as depicted in Table 4 in the paper. Although there are slight improvements in F1-OOS (2% score increase) for out-of-scope detection in most methods, these methods still underperform when recognizing known classes and in overall performance. Therefore, we opt for outlier exposure as a more effective technique to deal with out-of-scope samples and adopt this approach in our work.

Table 9: $K+1$ classification results on the MIntRec2.0 dataset.

| Methods | In-scope Classification | | | | | | In-scope + Out-of-scope Classification | | | |
| | F1 | P | R | ACC | WF1 | WP | F1-IS | ACC | F1-OOS | F1 |
|---|---|---|---|---|---|---|---|---|---|---|
| TEXT | 42.23 | 55.34 | 37.42 | 43.84 | 49.60 | 64.28 | 40.52 | 55.69 | 64.28 | 41.29 |
| MAG-BERT | 40.68 | 53.34 | 36.57 | 43.75 | 48.95 | 63.14 | 38.87 | 55.76 | 64.41 | 39.70 |
| MulT | 39.48 | 54.96 | 34.90 | 42.47 | 48.04 | 64.17 | 38.26 | 56.33 | 65.48 | 39.14 |
| Context TEXT | 40.33 | 50.45 | 36.97 | 43.72 | 47.80 | 59.18 | 38.21 | 54.65 | 63.79 | 39.04 |
| Context MAG-BERT | 43.14 | 53.20 | 39.34 | 47.09 | 51.70 | 62.53 | 40.87 | 55.65 | 64.04 | 41.62 |
| Context MulT | 42.46 | 54.72 | 38.28 | 31.54 | 35.80 | 65.88 | 40.38 | 42.59 | 50.02 | 40.69 |

## L    DATA SPLITS

We partition our dataset into training, validation, and testing sets at an approximate ratio of 7:1:1 for both utterances and dialogues. Detailed statistics for each set, encompassing both in-scope and out-of-scope data, are presented in Table 10.

Table 10:  Data splits of the MIntRec2.0 dataset. # denotes the number.

| Item | # Dialogues | # Utterances | # In-scope Utterances | # Out-of-scope Utterances |
|---|---|---|---|---|
| Total | 1,245 | 15,040 | 9,304 | 5,736 |
| Training | 871 | 9,989 | 6,165 | 3,824 |
| Validation | 125 | 1,821 | 1,106 | 715 |
| Testing | 249 | 3,230 | 2,033 | 1,197 |

## M    HYPER-PARAMETER CONFIGURATIONS

The comprehensive configurations of hyper-parameters used in our experiments are presented in Table 11, Table 12, Table 13, Table 14, Table 15, and Table 16.

Table 11:  The hyperparameters of the TEXT baseline in single-turn conversations.

| Setting | hyperparameters | value | Setting | hyperparameters | value |
|---|---|---|---|---|---|
| | *eval_monitor*: | *accuracy* | | *eval_monitor*: | *accuracy* |
| | *train_batch_size*: | 16 | | *train_batch_size*: | 16 |
| | *eval_batch_size*: | 8 | | *eval_batch_size*: | 8 |
| | *test_batch_size*: | 8 | | *test_batch_size*: | 8 |
| | *wait_patience*: | 8 | | *wait_patience*: | 8 |
| w / o OOS | *num_train_epochs*: | 40 | w OOS | *num_train_epochs*: | 40 |
| | *warmup_proportion*: | 0.1 | | *warmup_proportion*: | 0.1 |
| | *lr*: | 2e-5 | | *lr*: | 1e-5 |
| | *weight_decay*: | 0.1 | | *weight_decay*: | 0.1 |

## N    DIALOGUE INTENT CLASSIFICATION IN NLP

We have conducted experiments to benchmark our dataset with two state-of-the-art algorithms in open intent detection for NLP: DA-ADB (Zhang et al., 2023b) and KNNCL (Zhou et al., 2022) with

Table 12: The hyperparameters of the MAG-BERT baseline in single-turn conversations.

| Setting | hyperparameters | value | Setting | hyperparameters | value |
|---|---|---|---|---|---|
| w / o OOS | *need_aligned*: | *True* | w OOS | *need_aligned*: | *True* |
| | *eval_monitor*: | *accuracy* | | *eval_monitor*: | *accuracy* |
| | *train_batch_size*: | 16 | | *train_batch_size*: | 16 |
| | *eval_batch_size*: | 8 | | *eval_batch_size*: | 8 |
| | *test_batch_size*: | 8 | | *test_batch_size*: | 8 |
| | *wait_patience*: | 8 | | *wait_patience*: | 8 |
| | *num_train_epochs*: | 40 | | *num_train_epochs*: | 40 |
| | *beta_shift*: | 0.005 | | *beta_shift*: | 0.005 |
| | *dropout_prob*: | 0.5 | | *dropout_prob*: | 0.5 |
| | *warmup_proportion*: | 0.1 | | *warmup_proportion*: | 0.1 |
| | *lr*: | 5e-6 | | *lr*: | 5e-6 |
| | *aligned_method*: | *ctc* | | *aligned_method*: | *ctc* |
| | *weight_decay*: | 0.03 | | *weight_decay*: | 0.1 |

Table 13: The hyperparameters of the MulT baseline in single-turn conversations.

| Setting | hyperparameters | value | Setting | hyperparameters | value |
|---|---|---|---|---|---|
| w / o OOS | *padding_mode*: | *zero* | w OOS | *padding_mode*: | *zero* |
| | *padding_loc*: | *end* | | *padding_loc*: | *end* |
| | *need_aligned*: | *False* | | *need_aligned*: | *False* |
| | *eval_monitor*: | *accuracy* | | *eval_monitor*: | *accuracy* |
| | *train_batch_size*: | 16 | | *train_batch_size*: | 16 |
| | *eval_batch_size*: | 8 | | *eval_batch_size*: | 8 |
| | *test_batch_size*: | 8 | | *test_batch_size*: | 8 |
| | *wait_patience*: | 8 | | *wait_patience*: | 8 |
| | *num_train_epochs*: | 40 | | *num_train_epochs*: | 40 |
| | *dst_feature_dims* : | 80 | | *dst_feature_dims* : | 80 |
| | *nheads*: | 4 | | *nheads*: | 4 |
| | *n_levels*: | 8 | | *n_levels*: | 8 |
| | *attn_dropout*: | 0.0 | | *attn_dropout*: | 0.0 |
| | *attn_dropout_v*: | 0.1 | | *attn_dropout_v*: | 0.1 |
| | *attn_dropout_a*: | 0.1 | | *attn_dropout_a*: | 0.1 |
| | *relu_dropout*: | 0.3 | | *relu_dropout*: | 0.3 |
| | *embed_dropout*: | 0.0 | | *embed_dropout*: | 0.0 |
| | *res_dropout*: | 0.0 | | *res_dropout*: | 0.0 |
| | *output_dropout*: | 0.2 | | *output_dropout*: | 0.0 |
| | *text_dropout*: | 0.1 | | *text_dropout*: | 0.0 |
| | *grad_clip*: | 0.5 | | *grad_clip*: | 0.5 |
| | *attn_mask*: | *True* | | *attn_mask*: | *True* |
| | *conv1d_kernel_size_l*: | 5 | | *conv1d_kernel_size_l*: | 5 |
| | *conv1d_kernel_size_v*: | 1 | | *conv1d_kernel_size_v*: | 1 |
| | *conv1d_kernel_size_a*: | 1 | | *conv1d_kernel_size_a*: | 1 |
| | *lr*: | 5e-6 | | *lr*: | 5e-6 |

the open-source TEXTOIR platform (Zhang et al., 2021a). Consistent with the original settings of these algorithms, they are trained on in-scope samples and tested on both in-scope and out-of-scope samples. The results are shown in Table 17.

The results show that even state-of-the-art methods for open intent detection generally underperform compared to the BERT$_{\text{LARGE}}$ text classifier across most metrics. However, they do excel in identifying out-of-scope utterances, typically achieving higher F1-OOS scores. Notably, KNNCL also scores higher in accuracy.

## O  OUT-OF-DISTRIBUTION DETECTION ACROSS DIFFERENT SOURCES

We also explore the model performance in an out-of-distribution (OOD) setting across different sources. To address this, we have conducted experiments where we use data from one source as

Table 14: The hyperparameters of the TEXT baseline in multi-turn conversations.

| hyperparameters | value |
|---|---|
| *eval_monitor*: | *accuracy* |
| *train_batch_size*: | 2 |
| *eval_batch_size*: | 2 |
| *test_batch_size*: | 2 |
| *wait_patience*: | 3 |
| *num_train_epochs*: | 40 |
| *warmup_proportion*: | 0.1 |
| *lr*: | 1e-5 |
| *weight_decay*: | 0.1 |
| *train_batch_size*: | 16 |

Table 15: The hyperparameters of the MAG-BERT baseline in multi-turn conversations.

| hyperparameters | value |
|---|---|
| *need_aligned*: | *True* |
| *eval_monitor*: | *accuracy* |
| *train_batch_size*: | 2 |
| *select_batch_size*: | 16 |
| *eval_batch_size*: | 2 |
| *test_batch_size*: | 2 |
| *wait_patience*: | 3 |
| *num_train_epochs*: | 40 |
| *context_len*: | 0.5 |
| *beta_shift*: | 0.05 |
| *dropout_prob*: | 0.05 |
| *warmup_proportion*: | 0.01 |
| *lr*: | 4e-6 |
| *aligned_method*: | *conv1d* |
| *weight_decay*: | 0.1 |

the in-distribution dataset for training, validation, and testing. We then use data from the other two sources exclusively for OOD testing, in accordance with (Hendrycks & Gimpel, 2017; Liang et al., 2018). For evaluation, we utilize a comprehensive set of metrics: AUROC (Area Under the Receiver Operating Characteristic Curve), AUPR-In (Area Under the Precision-Recall Curve for in-distribution detection), AUPR-Out (Area Under the Precision-Recall Curve for OOD detection), FPR-95 (False Positive Rate at 95% True Positive Rate), and EER (Equal Error Rate). Higher scores are preferable for the first three metrics, while lower scores are desirable for the last two.

As shown in Table 18, the results indicate that MAG-BERT shows lower performance on OOD detection compared with the text baseline on most metrics. Both text and multimodal fusion methods achieve very low performance on OOD detection metrics, highlighting the substantial challenges presented by this setting. This opens up an intriguing avenue for future research in OOD detection under these conditions.

## P    CHATGPT PROMPTS

We provide prompts for both zero-shot (ChatGPT-0) and few-shot (ChatGPT-10) settings of Chat-GPT. The detailed prompts are as follows:

**ChatGPT-0 Prompts**: Here is a set of given intent labels: [ *Acknowledge*, *Advise*, *Agree*, *Apologise*, *Arrange*, *Ask for help*, *Asking for opinions*, *Care*, *Comfort*, *Complain*, *Confirm*, *Criticize*, *Doubt*, *Emphasize*, *Explain*, *Flaunt*, *Greet*, *Inform*, *Introduce*, *Invite*, *Joke*, *Leave*, *Oppose*, *Plan*, *Praise*, *Prevent*, *Refuse*, *Taunt*, *Thank*, *Warn*, *OOS*]. Additionally, *OOS* represents an unknown intent that does not belong to the known set of intents. Next, I will provide you with a collection of dialogs: *utterances*. The collection contains multiple utterances presented in sequential order, and they can be considered as contextualized conversations. When considering each sample and taking into account

Table 16: The hyperparameters of the MulT baseline in multi-turn conversations.

| hyperparameters | value |
|---|---|
| *padding_mode*: | *zero* |
| *padding_loc*: | *end* |
| *need_aligned*: | *False* |
| *eval_monitor*: | *accuracy* |
| *train_batch_size*: | 2 |
| *select_batch_size*: | 16 |
| *eval_batch_size*: | 2 |
| *test_batch_size*: | 2 |
| *wait_patience*: | 3 |
| *context_length*: | 1 |
| *num_train_epochs*: | 40 |
| *dst_feature_dims*: | 80 |
| *nheads*: | 4 |
| *n_levels*: | 8 |
| *attn_dropout*: | 0.0 |
| *attn_dropout_v*: | 0 |
| *attn_dropout_a*: | 0.1 |
| *relu_dropout*: | 0.2 |
| *embed_dropout*: | 0.1 |
| *res_dropout*: | 0 |
| *output_dropout*: | 0 |
| *text_dropout*: | 0.4 |
| *grad_clip*: | 0.5 |
| *attn_mask*: | *True* |
| *conv1d_kernel_size_l*: | 5 |
| *conv1d_kernel_size_v*: | 1 |
| *conv1d_kernel_size_a*: | 1 |
| *lr*: | 5e-6 |

Table 17: Performance of open intent detection on the MIntRec2.0 dataset.

| Methods | In-scope Classification | | | | | | Out-of-scope Classification | | | |
|---|---|---|---|---|---|---|---|---|---|---|
| | F1 | P | R | ACC | WF1 | WP | F1-IS | ACC | F1-OOS | F1 |
| TEXT | 51.60 | 55.47 | 51.31 | 59.30 | 58.01 | 58.85 | 43.37 | 43.24 | 30.40 | 42.96 |
| DA-ADB | 46.16 | 51.28 | 46.08 | 57.44 | 54.96 | 55.66 | 39.60 | 39.18 | 36.17 | 39.49 |
| KNNCL | 50.64 | 51.19 | 50.71 | 56.54 | 56.27 | 56.39 | 35.58 | 48.58 | 55.77 | 36.23 |

Table 18: OOD detection performance across different sources.

| ID source | OOD source(s) | Methods | AUROC | AUPR-In | AUPR-Out | FPR95 | EER |
|---|---|---|---|---|---|---|---|
| Superstore | Bigbang & Friends | TEXT | 51.33 | 21.75 | 80.25 | 93.47 | 49.43 |
| | | MAG-BERT | 50.96 | 21.28 | 80.14 | 93.74 | 49.21 |
| Bigbang | Superstore & Friends | TEXT | 51.33 | 21.75 | 80.25 | 93.47 | 49.43 |
| | | MAG-BERT | 50.96 | 21.28 | 80.14 | 93.74 | 49.21 |
| Friends | Bigbang & Superstore | TEXT | 55.97 | 26.17 | 80.56 | 91.22 | 45.40 |
| | | MAG-BERT | 51.01 | 25.62 | 79.81 | 92.57 | 45.90 |

its contextual information, please select an appropriate label from the intent label set (emphasis: you can only choose intent labels from the given set of intent labels). If there are no suitable labels in the set, assign the label of the sample as *OOS*. Please provide the output in the following format: *Serial number and original text of the sample: Intent label*. Apart from that, do not output anything else.

**ChatGPT-10 Prompts**: Here is a list of multiple multi-turn conversations. Each dictionary in the list represents a conversation paragraph, where each key-value pair represents an intent example

as a key and its corresponding label as a value. Next time I will enter my request, please only reply "received". This is a list of given intent labels: [ *Acknowledge*, *Advise*, *Agree*, *Apologise*, *Arrange*, *Ask for help*, *Asking for opinions*, *Care*, *Comfort*, *Complain*, *Confirm*, *Criticize*, *Doubt*, *Emphasize*, *Explain*, *Flaunt*, *Greet*, *Inform*, *Introduce*, *Invite*, *Joke*, *Leave*, *Oppose*, *Plan*, *Praise*, *Prevent*, *Refuse*, *Taunt*, *Thank*, *Warn*, *OOS*], where *OOS* represents an unknown intent that is not intended otherwise. Now, you need to learn from the conversations that you were given in the last Q&A, and then I'll provide you with a dialog that contains utterances in it, and these utterances are given in order and can be considered as contextual. Now, for each utterance that requires you to use the knowledge you gained from the given conversations, select a label as output from the given list of labels: for the following given dialog, in this format: *Original sample: Intent labels* output.

## Q    LIMITATIONS AND POTENTIAL NEGATIVE SOCIETAL IMPACTS

**Limitations:** This study presents several limitations that warrant acknowledgment. First, deploying this system in real-world settings necessitates collecting personal data, including facial expressions, voice, and text, thereby raising critical privacy concerns requiring meticulous attention. Second, the issue of liability remains ambiguous, especially in sensitive applications such as medical diagnosis, should the technology produce erroneous results. Third, our training dataset may lack comprehensive representation across diverse cultural backgrounds, potentially resulting in misunderstandings or the perpetuation of stereotypes. Lastly, substantial opportunities exist for enhancing the system's performance, particularly in effectively utilizing context information and out-of-scope sample data and incorporating non-verbal modalities.

**Potential Negative Societal Impacts:** While our work contributes valuable advancements in the field of multimodal intent recognition, it also has the potential to introduce negative societal impacts.

Firstly, there is the potential for misuse of our dataset if it becomes publicly available under an open-source license. Such misuse could include unauthorized commercial applications or other nefarious purposes that could result in harm. To mitigate this, we strongly urge users to adhere strictly to the licensing terms associated with this dataset.

Secondly, as AI systems like ours become increasingly sophisticated and prevalent, there is the risk of over-reliance on these technologies. This could lead to a decline in certain human skills, especially those related to understanding and interpreting conversational cues. As researchers and developers, we must continue to balance the advancement of AI with the preservation and enhancement of human capabilities.

Thirdly, the baseline system might be used with malicious intent. While any technology can be used for both beneficial and harmful purposes, our system is designed to detect out-of-scope (OOS) categories, which could be exploited to identify harmful or malicious intents. By integrating robust OOS detection, our system can flag conversations or utterances that deviate from predefined, acceptable intents. This feature could act as a first line of defense against technology misuse, as it can be tailored to detect and flag potentially harmful conversation intents.

Furthermore, establishing a benchmark in this field can have numerous positive societal impacts, such as enhancing human-computer interactions, aiding mental health assessments, and improving customer service automation. We believe the ethical deployment of this technology largely hinges on implementation safeguards and the specific contexts in which it is used.

