# OpenReview forum: "MIntRec2.0: A Large-scale Benchmark Dataset for Multimodal Intent Recognition and Out-of-scope Detection in Conversations"
_ICLR.cc/2024/Conference — ICLR 2024 poster_

### Official Review · Reviewer_ju3D · 2023-10-21

**Soundness:** 4 excellent
**Presentation:** 3 good
**Contribution:** 3 good
**Rating:** 6
**Confidence:** 4

**Summary:**

The authors identify two main gaps in current benchmarks for assessing multimodal intent recognition systems: multi-turn conversational interactions in the real world contain out-of-scope utterances (not relevant to the intent detection taxonomy), and the existence of multiple parties/agents in dialogs.
To address this gap, this paper proposes the MIntRec2.0 dataset with 1.2K conversations containing 15K annotated utterances harvested from three TV series totalling ~12 hours of dialogue.
The authors use strong LLM baselines (ChatGPT), human evaluation, and existing methods to populate the benchmark evaluation results and identify challenges in the dataset that are not addressed by existing models.

**Strengths:**

S1: The authors give detailed human performance results and identify the gap between SOTA multimodal approaches and humans. Furthermore, this indicates the benchmark is fairly difficult (not simple common sense) considering human performance of 71% with ~7% of training data.

S2: Comparison of resources in Table 1 is clear and convincing; in particular this dataset seems to be the first to include multi-party dialogs, and one of the only datasets with OOS labels. The expanded intent classes for the coarse-grained "Express Emotions" and "Achieve Goals" existing intents make sense and are sufficiently distinct from one another to add value to the taxonomy.

**Weaknesses:**

W1: Would have liked to see more discussion on the effect of incorporating multi-modal information aside from mentions of numbers and "indicating the challenge of using multi-modal information on out-of-scope data". A case study or deeper slice of results would be illuminating here, or even a deeper analysis in the main paper of what primarily constitutes OOS.

W2: More context should be provided on why 1-4% increases on metrics are considered significant in this case (is it statistically significant or is there some other meaning here e.g. for real life use cases?)

W3: Some clarity in the ChatGPT vs. Humans evaluation would be helpful - were the 10 dialogues of 227 utterances fixed across experiments, or were other few-shot training samples of 10 dialogs with different intent class balances attempted / were metrics aggregated? It would be helpful to picture whether the metric improvements are conditioned on specific intent classes.

**Questions:**

Q1: Given the TV series (comedy), how were things like laugh tracks or other "audience signals" of e.g. humor taken into account? What are the primary differences from the authors' perspective between these data sources and multi-turn conversations in the real world that intent detection systems would mainly work with?

Q2: Were annotators queried about what modalities of information were most valuable to their understanding of the conversational intent? This would be helpful information to gauge the differences between how humans and MMI systems work.

---

> ### Author Response · Authors · 2023-11-17
> **Response to Reviewer ju3D (Part 1/5)**
>
> Thanks for your suggestions. We will address your concerns point by point.
>
> **Reply to Weaknesses**:
>
> **A1**. Thank you for your insightful question. To address the impact of incorporating multimodal information in intent recognition, we first carried out additional experiments to examine the fine-grained performance of each class using both text-only and multimodal fusion baselines. We analyzed 30 specific intent classes and one out-of-scope category, employing the F1-score as the evaluation metric, akin to the approach in [1]. The average performances from 5 experimental runs are presented below:
>
> **a. Single-turn Conversations**:
> |Methods| Acknowledge | Advise | Agree | Apologise | Arrange | Ask for help | Ask for opinions | Care | Comfort | Complain |
> | --- | --- | --- | --- | --- | --- | --- | --- | --- | --- | --- |
> |Text| 49.51 | 55.03 | 50.45 | 88.96 | 48.34 | 57.97 | 47.89 | 45.98 | 43.55 | 31.02 |
> |MAG-BERT| **54.86** | 53.86 | **54.54** | **93.02** | **52.04** | **61.14** | **48.89** | **50.05** | 41.69 | **32.26** |
>
> |Methods| Confirm | Criticize | Doubt | Emphasize | Explain | Flaunt | Greet | Inform | Introduce | Invite |
> | --- | --- | --- | --- | --- | --- | --- | --- | --- | --- | --- |
> |Text| 46.38 | 33.84 | 52.38 | 1.10 | 46.43 | 13.41 | 74.59 | 40.83 | 30.26 | 27.85|
> |MAG-BERT| 45.68 | **37.24** | 50.68 | **1.18** | **47.80** | **23.70** | **77.09** | **41.25** | **31.80** | **47.36** |
>
> |Methods| Joke | Leave | Oppose | Plan | Praise | Prevent | Refuse | Taunt | Thank | Warn | OOS |
> | --- | --- | --- | --- | --- | --- | --- | --- | --- | --- | --- | --- |
> |Text | 3.14 | 47.26 | 56.88 | 51.43 | 61.03 | 52.41 | 20.64 | 12.61 | 91.12 | 18.97 | 30.40 |
> |MAG-BERT| **8.85** | **47.61** | 56.60 | **54.73** | **63.30** | 51.32 | **29.37** | **13.76** | **91.85** | **30.89** | **34.03** |
>
> Our results show significant enhancements in understanding **26** intent classes when integrating nonverbal modalities. Notably, **14 classes saw improvements of over 3 points**, including *Acknowledge, Agree, Apologise, Arrange, Ask for help, Care, Criticize, Flaunt, Invite, Joke, Plan, Refuse, Warn*, and *OOS*. These classes represent a mix of common and challenging scenarios, as well as out-of-scope instances, all requiring high-level cognitive inference and semantic understanding. Remarkably, we observed substantial improvements of over **10** points in challenging classes like *Flaunt* ($\uparrow$ 10.29), *Invite* ($\uparrow$ 19.51), and *Warn* ($\uparrow$ 11.92). This underscores the importance of nonverbal modalities in recognizing human intentions. While there are also classes that show lesser improvements or rely more on text modality, the performance with non-verbal modalities is competitive, demonstrating the substantial benefit they bring to intent recognition in multimodal scenarios.
>
> Moreover, we extended our research to multi-turn conversations, maintaining the same experimental settings as in single-turn conversations.
>
> **b. Multi-turn Conversations**:
>
> |Methods| Acknowledge | Advise | Agree | Apologise | Arrange | Ask for help | Ask for opinions | Care | Comfort | Complain |
> | --- | --- | --- | --- | --- | --- | --- | --- | --- | --- | --- |
> |Text| 55.04 | 55.28 | 51.43 | 91.65 | 48.00 | 56.94 | 45.71 | 55.76 | 48.98 | 40.30 |
> |MAG-BERT| **61.67** | **55.45** | **53.22** | **91.98** | **52.78** | **60.65** | **51.88** | **56.72** | **50.50** | **40.82** |
>
> |Methods| Confirm | Criticize | Doubt | Emphasize | Explain | Flaunt | Greet | Inform | Introduce | Invite |
> | --- | --- | --- | --- | --- | --- | --- | --- | --- | --- | --- |
> |Text| 47.66 | 36.32 | 49.89 | 2.86 | 47.28 | 15.46 | 78.15 | 43.40 | 32.85 | 39.58 |
> |MAG-BERT| 46.85 | **37.22** | **51.46** | 2.11 | **48.60** | 8.34 | **79.43** | **45.64** | 32.35 | 36.46 |
>
> |Methods| Joke | Leave | Oppose | Plan | Praise | Prevent | Refuse | Taunt | Thank | Warn | OOS |
> | --- | --- | --- | --- | --- | --- | --- | --- | --- | --- | --- | --- |
> |Text| 4.98 | 53.43 | 58.00 | 51.91 | 66.23 | 56.90 | 31.60 | 15.33 | 84.72 | 27.01 | 63.56 |
> |MAG-BERT| 4.66 | 52.01 | 55.87 | 48.44 | **67.65** | 46.67 | 21.36 | 13.27 | **85.23** | **28.19** | 62.52 |
>
> In these settings, MAG-BERT outperformed text-only modality in 18 classes. Specifically, it achieved improvements of over 3 points in four classes, including *Acknowledge, Arrange, Ask for help*, and *Ask for opinions*, and 1-2% improvements in 8 classes, such as *Agree, Comfort, Doubt, Explain, Greet, Inform, Praise*, and *Warn*. These classes encompass a significant portion of common interaction intents. However, the gains from nonverbal modalities in multi-turn conversations were not as pronounced as in single-turn conversations, indicating existing methodological limitations in handling out-of-scope utterances and fully utilizing context information. Addressing these challenges is crucial for future research and highlights both the importance and complexity of the MIntRec2.0 dataset.

---

> ### Author Response · Authors · 2023-11-17
> **Response to Reviewer ju3D (Part 2/5)**
>
> Second,  to further illustrate the impact of multimodal information in intent recognition, we selected a specific dialogue from our dataset for a detailed case study. This allowed us to compare the predictive accuracy of both text and MAG-BERT models for each utterance, considering speaker identity and ground truth. The results are as follows:
>
> |Index|Speaker|Utterance|Predicted Label / Confidence (TEXT)|Predicted Label / Confidence (MAG-BERT)| Ground Truth |
> | --- | --- | --- | --- | --- | --- |
> |0| Glenn |salvatore kazlauskas.|**Greet** / 0.0735| OOS / 0.8289 | OOS |
> |1| Nico | wait, you mean creepy sal? |Confirm / 0.9563| Confirm / 0.6321 | Confirm |
> |2| Glenn | the man is dead. |Inform / 0.4575| Inform / 0.3950 | Inform |
> |3| Dina | police said he's been dead for at least a year. |Inform / 0.9842| Inform / 0.6941 | Inform |
> |4| Amy | are you crying? |**Care** / 0.7637| **Doubt** / 0.3580 | Confirm |
> |5| Kelly | poor guy. |**Taunt** / 0.1573| **Taunt** / 0.1045 | OOS |
> |6| Amy | but you didn't know him. |**Comfort** / 0.0932| **Explain** / 0.1131 | OOS |
> |7| Kelly | but he was a human being. |**Emphasize** / 0.0693| **Introduce** / 0.0603 | OOS |
> |8| Cheyenne | when he looked at you, it felt like he was grabbing you. |**Complain** / 0.3858| **Complain** / 0.1209 | OOS |
> |9|Glenn | apparently he was doing some work behind the drywall outside the women's washroom and then his foot got caught in a beam, and he starved to death. |Inform / 0.3145| Inform / 0.5194 | Inform |
> |10|Glenn| we're not sure. |**Inform** / 0.1238| **Inform** / 0.1233| OOS |
> |11|Glenn| he drilled a hole into the women's washroom so... |Inform / 0.2779| Inform / 0.1861|Inform|
> |12|Glenn| why? |Doubt / 0.9722| Doubt / 0.8096 | Doubt |
> |13|Dina| i know we all assumed that was amy. |**Emphasize** / 0.0682| Explain / 0.0996 | Explain |
> |14|Amy| why--why me? |Doubt / 0.7256| Doubt / 0.6292 | Doubt |
> |15|Dina| cause, you know... |Explain / 0.9794| Explain / 0.4273 | Explain |
> |16|Dina| i'm sure there's a lot of churn going on in there. |**Comfort** / 0.1254| **Explain** / 0.0841 | Taunt |
> |17|Cheyenne| wait, so it's just going to sit in the store? |Doubt / 0.9335| Doubt / 0.8337 | Doubt |
> |18|Nico| uh...   i'm not working next to a dead body. |**Oppose** / 0.9767| **Oppose** / 0.3158 | Taunt|
> |19|Dina| technically we've all been working next sal's dead body for the past year. |**Explain** / 0.1855| **Explain** / 0.22875768 | Inform |
> |20|Dina| nobody complained until now. |**Emphasize** / 0.0534| **Inform** / 0.0770| OOS |
> |21|Jonah| that must've been sal's foot we found. |**Inform** / 0.078| **Introduce** / 0.1448 | OOS |
> |22|Dina| actually he still had both of his feet. |**Doubt** / 0.0581| **Inform** / 0.0656 | Oppose |
>
>
> The data reveals that MAG-BERT generally achieves high accuracy in most in-scope classes, except for some with complex semantics like Taunt, Oppose, and Confirm. Furthermore, in many correctly predicted in-scope classes, MAG-BERT demonstrates a high confidence level, often exceeding a 0.5 probability. In contrast, the TEXT model showed more errors than MAG-BERT in two utterances and tended to exhibit higher confidence in certain utterances (e.g., index 4, 5, 8, 18). This comparison highlights the effectiveness of incorporating non-verbal modalities over relying solely on textual information. However, it's important to note that MAG-BERT struggles with out-of-scope (OOS) utterances, often making errors in these categories. This observation suggests that while existing multimodal fusion methods have capabilities in recognizing known intents, their performance in detecting out-of-scope utterances is limited, pointing to a significant area for future research and development.
>
> Finally, we conducted an analysis of the composition of out-of-scope utterances. Upon examining a random sample of 100 OOS utterances, we identified several utterances that potentially belong to new intent classes with limited examples, such as *help*, *drive person away*, *guess*, and *wish*. We also noted that a significant portion of the sample (nearly 60%) consists of utterances that can be categorized as *statement-opinion*, where the speaker expresses personal views on a subject or individual. Another notable segment (nearly 20%) falls under *statement-non-opinion*, encompassing subjective statements. These findings correlate with the 42 dialogue act classes defined in the SwDA dataset [2] and align with our discussion in the main paper's Section 3, particularly in the subsection on out-of-scope utterances. Our study indicates the need for continuous exploration in this field to effectively identify and categorize a broader range of human intents, especially those that fall outside the currently defined scope.

---

> ### Author Response · Authors · 2023-11-17
> **Response to Reviewer ju3D (Part 3/5)**
>
> **A2**. Thank you for your question. I'll address the significance of the 1-4% improvements from both qualitative and quantitative perspectives:
>
> Firstly, from a qualitative standpoint, in the realm of existing multimodal language analysis tasks, state-of-the-art multimodal fusion methods typically exhibit modest improvements, generally ranging between 1-3 points, over previous baselines, such as in multimodal sentiment analysis [3, 4, 5, 6]. These incremental advancements are considered significant as they demonstrate the effectiveness and robustness of the proposed methods in real-world multimodal language datasets [7, 8]. The complexity of multimodal language analysis, necessitating nuanced modeling of cross-modal interactions, inherently makes significant performance enhancements challenging.
>
> Secondly, regarding quantitative results, we conducted statistical t-tests to compare the performance of multimodal fusion methods (MAG-BERT and MulT) against the text-only baseline (TEXT). The results demonstrate that the multimodal fusion methods significantly outperform the text baseline, with *p*-values < 0.05 (&dagger;) and < 0.1 (*), as detailed in the following tables. The *p*-values are provided in parentheses:
>
> |Methods| F1 | Precision | Recall | Accuracy | Weighted F1 | Weighted Precision |
> | --- | --- | --- | --- | --- | --- | --- |
> |TEXT|51.60|55.47|51.31|59.30|58.01|58.85|
> |MAG-BERT|55.17 (&dagger;, 0.0111) |57.78|55.10 (&dagger;, 0.0022)|60.58 (&dagger;, 0.0016)|59.68 (&dagger;, 0.0184)|59.98|
> |MulT|54.12 (&dagger;, 0.0412) |58.02|53.77 (&dagger;, 0.0228)|60.66 (&dagger;, 0.0047)|59.55 (&dagger;, 0.0194)|60.12|
>
> |Methods| In-scope F1 | Accuracy | Out-of-scope F1 | F1 |
> | --- | --- | --- | --- | --- |
> |TEXT|43.37|43.24|30.40|42.96|
> |MAG-BERT|46.48 (&dagger;, 0.0041) |44.80 (*, 0.0720)|34.03|46.08 (&dagger;, 0.0063)|
> |MulT|45.65 (&dagger;, 0.0164) |46.14|38.57|45.42 (&dagger;, 0.0172)|
>
> These results indicate that the multimodal fusion methods significantly outperform the text baseline in 4 out of 6 metrics for in-scope classification (F1, Recall, Accuracy, Weighted F1) and in 3 out of 4 metrics for combined in-scope and out-of-scope classification (In-scope F1, Accuracy, F1). While other metrics such as Precision, Weighted Precision, and Out-of-scope F1 show more variability, they also demonstrate substantial average improvements across five experimental runs. These findings validate the effectiveness of our methods and highlight the significance of the observed improvements across most metrics.
>
> **A3**. Thank you for your insightful question. In our experiments comparing ChatGPT and humans, the set of 10 dialogues totaling 227 utterances remained consistent across all experiments to ensure a fair comparison between ChatGPT and human participants. The performance metrics for each method were aggregated across all intent classes. To delve deeper into the performance on specific intent classes, we conducted additional experiments for ChatGPT-10, Humans-10, and Humans-100. We calculated the F1-score for each intent class and have presented these results as follows:
>
> |Methods| Acknowledge | Advise | Agree | Apologise | Arrange | Ask for help | Ask for opinions | Care | Comfort | Complain |
> | --- | --- | --- | --- | --- | --- | --- | --- | --- | --- | --- |
> |ChatGPT-10| 24.44 | 30.84 | 35.29 | 62.94 | 17.39 | 28.39 | 22.95 | 0.00 | 40.00 | 34.16 |
> |Humans-10|46.27 | 67.76 | 61.90 | 93.02 | 55.46 | 69.33 | 50.00 | 50.57 | 66.67 | 49.54 |
> |Humans-100| **69.23** | **70.73** | **65.67** | **93.02** | **64.22** | **79.45** | **61.11** | **62.34** | **77.23** | **64.11** |
>
> |Methods| Confirm | Criticize | Doubt | Emphasize | Explain | Flaunt | Greet | Inform | Introduce | Invite |
> | --- | --- | --- | --- | --- | --- | --- | --- | --- | --- | --- |
> |ChatGPT-10|25.00 | 14.12 | 15.05 | 6.45 | 33.55 | 18.46 | 64.52 | 31.76 | 8.22 | 47.06 |
> |Humans-10|60.19 | 59.05 | 60.67 | 31.03 | 55.67 | 40.00 | 86.13 | 50.41 | 47.76 | 38.71 |
> |Humans-100|**62.43** | **67.96** | **68.35** | **42.86** | **65.79** | **67.92** | **90.91** | **67.09** | **66.67** | **70.97** |
>
> |Methods| Joke | Leave | Oppose | Plan | Praise | Prevent | Refuse | Taunt | Thank | Warn | OOS |
> | --- | --- | --- | --- | --- | --- | --- | --- | --- | --- | --- | --- |
> |ChatGPT-10|16.00 | 33.33 | 25.00 | 29.27 | 50.00 | 12.12 | 16.87 | 13.48 | 59.09 | 37.04 | 27.85 |
> |Humans-10|28.17 | 71.30 | 67.51 | 57.78 | 73.06 | 70.77 | 47.83 | 44.62 | 93.91 | 34.78 | 62.83 |
> |Humans-100| **50.00** | **79.28** | **67.56** | **70.89** | **79.65** | **77.97** | **53.66** | **63.93** | **94.83** | **62.86** | **75.41** |

---

> ### Author Response · Authors · 2023-11-17
> **Response to Reviewer ju3D (Part 4/5)**
>
> The results illustrate that humans significantly outperform ChatGPT. With the same foundational knowledge of 10 dialogues encompassing 227 utterances, Humans-10 exhibited superior performance across nearly all intent classes and the out-of-scope category, outperforming ChatGPT-10 by over 10 points in most cases. **Notably, 15 intent classes and one out-of-scope category saw improvements of over 30 points. Classes like *Care* and *Prevent* achieved improvements of over 50 points, and *Ask for help*, *Criticize*, *Doubt*, and *Oppose* saw over 40 points improvement.** These findings highlight a significant gap between ChatGPT and human capabilities, underscoring humans' adeptness at using limited prior knowledge from multimodal contexts, such as body language and facial expressions, to infer and synthesize complex intents at a cognitive level—a skill where current machine learning methods, including large language models, fall short.
>
> Additionally, with more extensive prior knowledge of 100 dialogues comprising 997 utterances, Humans-100 performed even better compared to Humans-10 and achieved state-of-the-art performance across all classes. **This included markedly improved performance of over 10 points in 16 intent classes and the out-of-scope category**. This demonstrates the remarkable potential of humans to leverage multimodal knowledge and their ability to learn effectively with only a marginally larger dataset (7% of all training data). This proficiency even surpasses current fully supervised multimodal fusion methods, as shown in Table 4. The detailed intent performance comparison between ChatGPT and humans further validates the challenges presented by the MIntRec2.0 dataset, indicating that there is still considerable progress to be made in AI for the complex task of multimodal intent recognition. We plan to include these results in the Appendix of the revised paper.
>
>
> **Reply to Questions**:
>
> **A1**. Thank you for your question. In the process of data collection and dialogue segmentation from the TV series, we indeed encountered laugh tracks and other audience signals following the utterances. To ensure data integrity and focus on the complexity of the intents, we meticulously annotated only the speaking parts, excluding these external audience cues. This approach helps maintain data quality by providing relatively pure multimodal signals for analysis.
>
> Regarding the TV series as a data source, they are derived from real-world, open-ended, multi-turn conversational scenarios with a diverse range of scenes and topics (as detailed in our response to Reviewer SFq3-Reply to Weaknesses-A1). This makes them substantially similar to multi-turn conversations in real-world contexts, offering a rich resource for gathering human intents from everyday interactions.
>
> However, a primary distinction lies in the scope of intent categories defined in our study. The 30 intent categories we identified are based on existing hierarchical multimodal intent taxonomies [1], encompassing 16 classes in *Express Emotions or Attitudes* and 14 classes in *Achieve Goals*. These categories, which are prevalent in daily life, were determined after extensive analysis and summarization of dialogue videos. They theoretically align with philosophical bases of intent, encompassing both the traditional artificial intelligence definitions of intent (as plans or goals of an agent, coupled with corresponding feedback actions [9, 10]) and the emotional evaluation factors influenced by the brain [11]. However, we recognize that some real-world intent classes may not be fully covered by our defined categories. To address this, we included an additional out-of-scope (OOS) category for utterances that do not fit into any known classes. This category facilitates the exploration of new potential intents and, with further real-world data, can aid in broadening the scope of intent classification beyond a closed-world framework.
>
> In summary, our intent detection system is designed to be adaptable to real-world scenarios, and the chosen data sources provide a solid foundation for future research into human intent recognition.

---

> ### Author Response · Authors · 2023-11-17
> **Response to Reviewer ju3D (Part 5/5)**
>
> **A2**. Thank you for raising this important question. Firstly, it's crucial to note that intent recognition primarily stems from natural language processing, with significant foundational research and advancements in this area [12, 13, 14], including high performance in some goal-oriented dialogue systems. As a result, the text modality often plays a central role in deciphering complex human intentions, as evidenced in [1]. Nevertheless, in real-world settings, the integration of non-verbal modalities, such as video and audio, along with conversational context and scene information, is crucial to accurately infer intentions. Recognizing the limitations of existing datasets, which are often small-scale, single-turn, and limited to closed-world classification, our MIntRec2.0 dataset aims to facilitate research into the effectiveness of non-verbal modalities in multi-turn conversations and in contexts that more closely resemble real-world situations, including out-of-scope utterances.
>
> In the process of multimodal intent annotation, our annotators utilized textual, video, and audio information, combined with contextual data within each dialogue, to analyze the underlying intention of each utterance. To investigate the relative importance of different modalities in understanding conversational intent, we gathered insights on key aspects including text modality (*spoken language*), video modality (*facial expressions* and *body language*), audio modality (*tone*), and background (*context information* and *conversational scenes*). We asked 9 annotators involved in both annotation and human evaluation to rank these six aspects based on their importance in understanding human intentions, drawing from their experience. The aggregated ranking results are as follows:
>
> |Rank|Spoken Language| Facial Expressions | Tone | Context Information | Body Language | Conversational Scenes |
> | --- | --- | --- | --- | --- | --- | --- |
> |1| **6** | 0 | 2 | 1 | 0 | 0 |
> |2| 2 | **4** | 2 | 1 | 0 | 0 |
> |3| 1 | 3 | **3** | 1 | 1 | 0 |
> |4| 0 | 2 | 1 | **4** | 2 | 0 |
> |5| 0 | 0 | 0 | 2 |  **6**| 1 |
> |6| 0 | 0 | 0 | 0 | 1 | **8** |
>
> These results suggest that *spoken language* is the most critical factor, followed by *facial expressions, tone, context information, body language*, and *conversational scenes*. While *spoken language* is predominant, non-verbal cues like *facial expressions* and *tone* are valuable in perceiving emotions or attitudes, especially in the *Express Emotions or Attitudes* category. *Context information* provides essential background knowledge about the speakers, aiding in a more profound understanding of intentions. *Body language*, though complex and implicit, is also insightful, particularly in the *Achieve Goals* category. Conversational scenes, while less critical in open-ended dialogues, still contribute to understanding intent in specific contexts. We plan to include these insights and discussions in the Appendix of the revised paper to provide a more comprehensive understanding of the multimodal intent recognition process.
>
> **References**:
>
> [1] MIntRec: A New Dataset for Multimodal Intent Recognition. Zhang et al. ACM MM 2022.
> [2] Switchboard: Telephone speech corpus for research and development. Godfrey et al. ICASSP. 1992.
> [3] MISA: Modality-Invariant and -Specific Representations for Multimodal Sentiment Analysis. Hazarika et al. ACM MM 2022.
> [4] Integrating Multimodal Information in Large Pretrained Transformers. Rahman et al. ACL 2020.
> [5] Learning Modality-Specific Representations with Self-Supervised Multi-Task Learning for Multimodal Sentiment Analysis. AAAI 2021.
> [6] Improving Multimodal Fusion with Hierarchical Mutual Information Maximization for Multimodal Sentiment Analysis. Han et al. EMNLP 2021.
> [7] MOSI: Multimodal Corpus of Sentiment Intensity and Subjectivity Analysis in Online Opinion Videos. Zadeh et al. arXiv. 2016.
> [8] Multimodal Language Analysis in the Wild: CMU-MOSEI Dataset and Interpretable Dynamic Fusion Graph. Zadeh et al. ACL 2018.
> [9] Intention,–Plans,–and–Practical–Reason. Michael E Bratman. Mind. 1988.
> [10] Intelligent agents: Theory and practice. Michael Wooldridge and Nicholas R Jennings. The knowledge engineering review. 1995.
> [11] Intention, emotion, and action: A neural theory based on semantic pointers. Schröder et al. Cognitive science. 2014.
> [12] Snips Voice Platform: An Embedded Spoken Language Understanding System for Private-by-design Voice interfaces. Coucke et al. arXiv. 2018.
> [13] An evaluation dataset for intent classification and out-of-scope prediction. Larson et al. EMNLP-IJCNLP 2019.
> [14] Efficient Intent Detection with Dual Sentence Encoders. Casanueva et al. arXiv. 2020.

---

> > ### Comment · Reviewer_ju3D · 2023-11-22
> >
> > Thanks for the thorough responses to my questions. The answers to W1 in mentioning confidence does raise a question about whether model scores are properly calibrated, which is not discussed but perhaps *should* be. Nonetheless, I have raised my soundness score to 4.

---

### Official Review · Reviewer_kGfg · 2023-10-28

**Soundness:** 3 good
**Presentation:** 4 excellent
**Contribution:** 3 good
**Rating:** 8
**Confidence:** 3

**Summary:**

This paper proposes MIntRec2.0, a large-scale multimodal multi-party benchmark dataset that comprises 1,245 high-quality dialogues, totaling 12.3 hours, which is interesting and valuable for multimodal training and evaluation. The proposed dataset serves as a valuable
resource, providing a pioneering foundation for research in human-machine conversational interactions, and significantly facilitating related applications.

**Strengths:**

The proposed dataset is interesting and valuable for research in human-machine conversational interactions. The motivation is clear, the authors proposed three limitation including single-turn utterances, scales and out-of-scope utterances. The overall structure is well organised. In addition to more than 9,300 in-scope samples, it also includes over 5,700 out-of-scope samples appearing in multi-turn contexts, which naturally occur in real-world open scenarios, enhancing its practical applicability. Furthermore, they provide comprehensive information on the speakers in each utterance, enriching its utility for multi-party conversational research. This paper is a good dataset and resource paper.

**Weaknesses:**

no obvious flaws. The figures should be expanded.

**Questions:**

None

---

> ### Author Response · Authors · 2023-11-21
> **Response to Reviewer kGfg**
>
> We are greatly appreciate for your recognition of our contributions in this work.
>
> Thanks again for your positive evaluations!

---

### Official Review · Reviewer_SFq3 · 2023-11-01

**Soundness:** 3 good
**Presentation:** 3 good
**Contribution:** 3 good
**Rating:** 6
**Confidence:** 4

**Summary:**

Multimodal intent recognition is very important to natural human-computer interactions and has gained more attention in recent years. The authors released a new version of MIntRec, named MIntRec 2.0, which contains more categories and considers out-of-scope scenarios. It will support more explorations in this field.
There are also some limitations of this work, the authors should add more experiments to support the effectiveness of this dataset and improve the writing to make the paper more clear.

**Strengths:**

This work builds a larger multimodal intent recognition dataset under interaction scences with 30 categories of fine-grained intent annotations and some out-of-scope samples, which is closer to the real-world scenarios.

**Weaknesses:**

1. The dataset is only from three different TV series, which limits the diversity of scenes and topics.
2. The multimodal fusion performance is not obvious in some metrics, the authors should explain the results more clearly, which can support the effectiveness of the multimodal intention dataset.
3. There are also some mirror errors, such as:
1) A representative sample is depicted in Figure 5.  -> Figure 1
2) Interpretations of both the expanded and existing intent categories can be found in Table 7 and Appendix G, respectively. -> Table 2

**Questions:**

1. What do you think are the reasons why the improvement of using non-verbal multimodality is not obvious?
2. The intention of a speaker in interactions is very difficult and complex, so what do you think are the influencing factors?
3. What do you think is the difference between a human-computer interaction dataset and the proposed human-human interaction dataset for the multimodal intent classification task?

---

> ### Author Response · Authors · 2023-11-14
> **Response to Reviewer SFq3 (Part 1/4)**
>
> Thanks for your suggestions. We will address your concerns point by point.
>
> **Reply to Weaknesses**:
> **A1**. Thank you for your valuable feedback regarding the diversity of scenes and topics in our dataset. The first multimodal intent recognition dataset, MIntRec [1], is based on the TV series Superstore, featuring a rich array of characters and a multitude of storylines set in various scenes like shopping malls, warehouses, and offices. This dataset also presents hierarchical intent taxonomies with two broad categories, *Express Emotions or Attitudes* and *Achieve Goals*, which are grounded in philosophical theories [2], and delineates 20 fine-grained classes. Nonetheless, its data scale is somewhat limited.
>
> To address this and provide a more comprehensive resource for multimodal intent research, we introduced MIntRec2.0. This enhanced version incorporates two additional popular TV series, Friends and The Big Bang Theory. With 34 main characters and more than 10 primary types of conversational scenes and 15 distinct topics, MIntRec2.0 significantly enriches the dataset in terms of content. It covers a wide range of common intents encountered in daily life and expands the existing 20 classes to 30. Specifically, these conversational scenes and topics are diverse and include:
>
> **a. Scenes and Settings**:
> *Superstore* provides a unique retail environment with scenes in the store, cash registers, break room, parking lot, managerial offices, and warehouse, reflecting workplace dynamics and customer interactions.
> *Friends* showcases diverse social settings like the Central Perk, apartments, travel locations, and various city spots, emphasizing personal and relational interactions.
> *The Big Bang Theory* offers academic and living spaces, including apartments and the university, highlighting intellectual and social engagements.
>
> Each of these series brings a unique set of environments and interaction dynamics, ranging from personal and intimate to professional and public. The diversity in character backgrounds, professions, and social settings across these shows ensures a wide-ranging exploration of human interactions and conversational intents.
>
> Moreover, these series are culturally iconic and have significantly influenced societal communication patterns, making them highly relevant for studying contemporary conversational trends. Their popularity also ensures that the dataset is relatable and accessible for a broad range of researchers and applications.
>
> **b. Topics and Themes**:
> *Superstore* touches on workplace relations, management challenges, customer service scenarios, labor issues, and social issues like immigration and corporate dynamics.
> *Friends* explores friendship, romantic relationships, career challenges, and urban living, offering insights into a variety of emotional and relational topics.
> *The Big Bang Theory* delves into scientific discourse, geek culture, technological advancements, scientific research, social awkwardness, and the balance between intellectual pursuits and everyday life.
>
> The combination of these series presents an extensive range of human experiences and topics, from the mundane to the complex. This diversity enriches our dataset, making it an invaluable tool for studying and understanding the nuances of multimodal intent recognition in varied conversational contexts.
>
> Furthermore, these series, with their wide cultural impact, provide a relatable and realistic reflection of contemporary social dynamics, essential for developing robust and applicable AI models in the field of human-computer interaction.
>
> In summary, the chosen TV series offer a balanced mix of scenes and topics, providing a comprehensive resource that captures the complexity of human interactions and conversational intents. We are confident that our dataset's scope and diversity significantly contribute to the advancement of multimodal intent recognition research.
>
> **A2**. Thank you for your insightful feedback. In our experiments, incorporating nonverbal modalities with multimodal fusion methods resulted in noticeable performance improvements. Specifically, we observed improvements of 1-4% in in-scope classification and 1-8% in both in-scope and out-of-scope utterance identification in single-turn conversations. To further demonstrate the effectiveness of nonverbal modalities, we conducted experiments across 30 specific intent classes and one out-of-scope category, using the F1-score as the evaluation metric, similar to [1]. The average performance over 5 experimental runs is presented below:

---

> ### Author Response · Authors · 2023-11-14
> **Response to Reviewer SFq3 (Part 2/4)**
>
> **a. Single-turn Conversations**:
> |Methods| Acknowledge | Advise | Agree | Apologise | Arrange | Ask for help | Ask for opinions | Care | Comfort | Complain |
> | --- | --- | --- | --- | --- | --- | --- | --- | --- | --- | --- |
> |Text| 49.51 | 55.03 | 50.45 | 88.96 | 48.34 | 57.97 | 47.89 | 45.98 | 43.55 | 31.02 |
> |MAG-BERT| **54.86** | 53.86 | **54.54** | **93.02** | **52.04** | **61.14** | **48.89** | **50.05** | 41.69 | **32.26** |
>
> |Methods| Confirm | Criticize | Doubt | Emphasize | Explain | Flaunt | Greet | Inform | Introduce | Invite |
> | --- | --- | --- | --- | --- | --- | --- | --- | --- | --- | --- |
> |Text| 46.38 | 33.84 | 52.38 | 1.10 | 46.43 | 13.41 | 74.59 | 40.83 | 30.26 | 27.85|
> |MAG-BERT| 45.68 | **37.24** | 50.68 | **1.18** | **47.80** | **23.70** | **77.09** | **41.25** | **31.80** | **47.36** |
>
> |Methods| Joke | Leave | Oppose | Plan | Praise | Prevent | Refuse | Taunt | Thank | Warn | OOS |
> | --- | --- | --- | --- | --- | --- | --- | --- | --- | --- | --- | --- |
> |Text | 3.14 | 47.26 | 56.88 | 51.43 | 61.03 | 52.41 | 20.64 | 12.61 | 91.12 | 18.97 | 30.40 |
> |MAG-BERT| **8.85** | **47.61** | 56.60 | **54.73** | **63.30** | 51.32 | **29.37** | **13.76** | **91.85** | **30.89** | **34.03** |
>
> Our results show significant enhancements in understanding **26** intent classes when integrating nonverbal modalities. Notably, **14 classes saw improvements of over 3 points**, including *Acknowledge, Agree, Apologise, Arrange, Ask for help, Care, Criticize, Flaunt, Invite, Joke, Plan, Refuse, Warn*, and *OOS*. These classes represent a mix of common and challenging scenarios, as well as out-of-scope instances, all requiring high-level cognitive inference and semantic understanding. Remarkably, we observed substantial improvements of over **10** points in challenging classes like *Flaunt* ($\uparrow$ 10.29), *Invite* ($\uparrow$ 19.51), and *Warn* ($\uparrow$ 11.92). This underscores the importance of nonverbal modalities in recognizing human intentions. While there are also classes that show lesser improvements or rely more on text modality, the performance with non-verbal modalities is competitive, demonstrating the substantial benefit they bring to intent recognition in multimodal scenarios.
>
> Moreover, we extended our research to multi-turn conversations, maintaining the same experimental settings as in single-turn conversations.
>
> **b. Multi-turn Conversations**:
>
> |Methods| Acknowledge | Advise | Agree | Apologise | Arrange | Ask for help | Ask for opinions | Care | Comfort | Complain |
> | --- | --- | --- | --- | --- | --- | --- | --- | --- | --- | --- |
> |Text| 55.04 | 55.28 | 51.43 | 91.65 | 48.00 | 56.94 | 45.71 | 55.76 | 48.98 | 40.30 |
> |MAG-BERT| **61.67** | **55.45** | **53.22** | **91.98** | **52.78** | **60.65** | **51.88** | **56.72** | **50.50** | **40.82** |
>
> |Methods| Confirm | Criticize | Doubt | Emphasize | Explain | Flaunt | Greet | Inform | Introduce | Invite |
> | --- | --- | --- | --- | --- | --- | --- | --- | --- | --- | --- |
> |Text| 47.66 | 36.32 | 49.89 | 2.86 | 47.28 | 15.46 | 78.15 | 43.40 | 32.85 | 39.58 |
> |MAG-BERT| 46.85 | **37.22** | **51.46** | 2.11 | **48.60** | 8.34 | **79.43** | **45.64** | 32.35 | 36.46 |
>
> |Methods| Joke | Leave | Oppose | Plan | Praise | Prevent | Refuse | Taunt | Thank | Warn | OOS |
> | --- | --- | --- | --- | --- | --- | --- | --- | --- | --- | --- | --- |
> |Text| 4.98 | 53.43 | 58.00 | 51.91 | 66.23 | 56.90 | 31.60 | 15.33 | 84.72 | 27.01 | 63.56 |
> |MAG-BERT| 4.66 | 52.01 | 55.87 | 48.44 | **67.65** | 46.67 | 21.36 | 13.27 | **85.23** | **28.19** | 62.52 |
>
> In these settings, MAG-BERT outperformed text-only modality in 18 classes. Specifically, it achieved improvements of over 3 points in four classes, including *Acknowledge, Arrange, Ask for help*, and *Ask for opinions*, and 1-2% improvements in 8 classes, such as *Agree, Comfort, Doubt, Explain, Greet, Inform, Praise*, and *Warn*. These classes encompass a significant portion of common interaction intents. However, the gains from nonverbal modalities in multi-turn conversations were not as pronounced as in single-turn conversations, indicating existing methodological limitations in handling out-of-scope utterances and fully utilizing context information. Addressing these challenges is crucial for future research and highlights both the importance and complexity of the MIntRec2.0 dataset. We would like to include these experimental results and discussions in the appendix of the revised paper.
>
>
> **A3**. We appreciate your attention to detail and will ensure that all the identified errors are corrected in the revised version of the paper.

---

> ### Author Response · Authors · 2023-11-14
> **Response to Reviewer SFq3 (Part 3/4)**
>
> **Reply to Questions**:
>
> **A1**. Thank you for your insightful feedback.
>
> Firstly, intent recognition has its roots in natural language processing, where numerous foundational works have focused on understanding the semantics of utterances, particularly within goal-oriented dialogue systems [3, 4, 5]. Consequently, it's unsurprising that text modality often serves as the primary means for interpreting multimodal language. This predominance of text is further evidenced in other well-known multimodal tasks like sentiment analysis and emotion recognition [6, 7, 8, 9, 10]. However, in real-world scenarios, the integration of non-verbal modalities becomes essential. Our world is inherently multimodal, and non-verbal cues significantly enhance the comprehension of true intentions, especially in open-ended dialogues. Recognizing the need for a high-quality, large-scale multimodal intent dataset, we developed MIntRec2.0 to bridge this gap and facilitate further research.
>
> Secondly, the incorporation of non-verbal modalities indeed yields notable improvements. As indicated in Table 4 of our main paper, multimodal fusion methods such as MAG-BERT and MulT demonstrate score enhancements of 1-4% across various metrics in both in-scope classification and out-of-scope detection. While these modalities might not always yield substantial improvements in certain aspects like handling out-of-scope utterances or contextual information, this is largely due to the absence of fusion methods specifically designed for these purposes. However, as detailed in **Reply to Weaknesses**-A2, MAG-BERT notably enhances performance across many intent classes, highlighting the significant potential of non-verbal modalities.
>
> Finally, the MIntRec2.0 dataset presents the first large-scale resource in multimodal intent research, offering a comprehensive framework for benchmarking with advanced multimodal fusion methods. Our extensive evaluations reveal that while non-verbal modalities contribute to recognizing complex cognitive-level intents, they face challenges in effectively leveraging context and out-of-scope utterances. This limits their robustness and generalizability in this demanding task. Comparing the performances of humans and ChatGPT, we observe that humans excel at using limited multimodal language as prior knowledge for synthesizing and inferring real intentions, significantly outperforming even sophisticated large language models. This distinction underscores the importance and challenges posed by this dataset, marking it as a valuable contribution to the research community.
>
> **A2**. In addressing the complexity of understanding a speaker's intentions during interactions, two main aspects emerge: non-verbal information and background information. The first aspect involves effectively leveraging non-verbal modalities, such as video and audio, to capture cross-modal interactions. The second aspect encompasses various types of side information, including conversational context, out-of-scope utterances, and other objective factors (e.g., cultural and social background, scenes, and topics). These elements serve as implicit prior knowledge, aiding in the understanding of human intentions. A detailed introduction of these perspectives is as follows:
>
> * **Non-Verbal Cues**: Non-verbal modalities such as video and audio are also significant for understanding human intentions in multimodal scenarios. Tones, gestures, facial expressions, body language, and eye contact are crucial in providing additional information about the intention of a speaker. These cues can add to, contradict, or enhance the verbal information, offering deeper insights into the speaker's real intentions.
> * **Background Information**:
>     - **Conversational Context**: This refers to the specific flow of information within the dialogue and the relevance of previous utterances to the current one. Understanding the conversational history is crucial for interpreting the speaker's current intentions. The context may include the topics being discussed, the progress of the dialogue interactions, and the shared knowledge and experiences between the speakers, which are significant in helping understand intentions.
>     - **Out-of-scope Utterances**: The occurrence of statements or new intentions that deviate from the known intentions of the speaker can help discover new intents. On the one hand, understanding out-of-scope utterances helps avoid misclassifying them as in-scope intents. On the other hand, detected out-of-scope utterances (e.g., unrelated or new topics) can be used to understand broader communicative goals, facilitate human communication, and identify potential communication interests.

---

> ### Author Response · Authors · 2023-11-15
> **Response to Reviewer SFq3 (Part 4/4)**
>
> * **Background Information**:
>     - **Objective Factors**: (1) Cultural and Social Background: The cultural norms, social backgrounds, and linguistic styles of the speakers influence how intentions are expressed and perceived. This includes factors like accent, dialect, and the social context of the conversation. (2) Scenes and Topics: The physical setting of the conversation and the subject matter can have a significant impact on a speaker's intent. Different environments and topics can elicit varied communicative intents and styles.
>
> In developing the MIntRec2.0 dataset, we have diligently incorporated these key factors to facilitate a more accurate understanding of human intentions. Firstly, it contains 1,245 dialogues, each with a conversational context. Secondly, of the total 15,040 utterances, each contains both verbal and nonverbal (video and audio) information. Thirdly, we consider out-of-scope utterances that typically occur in natural interactions, annotating not only 30 in-scope intent categories but also one out-of-scope category, resulting in 9,304 in-scope and 5,736 out-of-scope utterances. Finally, we annotate the speaker identity for each utterance to help researchers distinguish the different objective factors of various speakers. Therefore, we believe MIntRec2.0 is a foundational and significant contribution to research in human intention understanding.
>
> **A3**. The difference between human-computer and human-human interaction datasets is substantial, primarily due to the inherent differences in the nature of communication and interaction in each setting. Here are the primary three distinctions:
> * **Interaction Dynamics**: In human-computer interactions, the dynamics are typically unidirectional or asymmetrical. Users often lead the conversational directions and give commands to generate dialogue utterances (e.g., in goal-oriented dialogue systems [3, 4]). In contrast, human-human interactions are more dynamic and bidirectional (e.g., in open-ended dialogue systems [11]), with both parties actively contributing, responding, and adapting to the conversation flow.
> * **Complexity of Communication**: Human-computer interactions are generally more structured and predictable, with a limited range of intents that follow specific orders or needs and relatively simple responses. Human-human interactions are far more complex, involving a wider range of intents, subtleties, emotions, and unpredictability.
> * **Non-Verbal Cues**: Non-verbal cues are often limited or absent in human-computer interactions, as seen in many task-oriented datasets in NLP [3, 4, 5]. Even with advanced multimodal fusion algorithms, interpreting non-verbal cues from humans remains challenging for computers. In human-human interactions, non-verbal cues play a crucial role in understanding intent, with nuances in body language, facial expressions, and tone carrying significant information.
>
> Therefore, constructing a human-human interaction dataset for multimodal intent recognition is more appropriate, as it satisfies interaction dynamics, supports complex communication, and incorporates non-verbal cues. MIntRec2.0 makes a pioneering contribution in this area and aims to facilitate related research and application.
>
> **References**:
> [1] MIntRec: A New Dataset for Multimodal Intent Recognition. Zhang et al. ACM MM 2022.
> [2] Intention, emotion, and action: A neural theory based on semantic pointers. Schröder et al. Cognitive science. 2014.
> [3] Snips Voice Platform: An Embedded Spoken Language Understanding System for Private-by-design Voice interfaces. Coucke et al. arXiv. 2018.
> [4] An evaluation dataset for intent classification and out-of-scope prediction. Larson et al. EMNLP-IJCNLP 2019.
> [5] Efficient Intent Detection with Dual Sentence Encoders. Casanueva et al. arXiv. 2020.
> [6] Learning Modality-Specific Representations with Self-Supervised Multi-Task Learning for Multimodal Sentiment Analysis. AAAI 2021.
> [7] Improving Multimodal Fusion with Hierarchical Mutual Information Maximization for Multimodal Sentiment Analysis. Han et al. EMNLP 2021.
> [8] MOSI: Multimodal Corpus of Sentiment Intensity and Subjectivity Analysis in Online Opinion Videos. Zadeh et al. arXiv. 2016.
> [9] Multimodal Language Analysis in the Wild: CMU-MOSEI Dataset and Interpretable Dynamic Fusion Graph. Zadeh et al. ACL 2018.
> [10] MELD: A Multimodal Multi-Party Dataset for Emotion Recognition in Conversations. Poria et al. ACL 2019.
> [11] IEMOCAP: interactive emotional dyadic motion capture database. Busso et al. Language resources and evaluation. 2008.

---

### Official Review · Reviewer_Un9C · 2023-11-13

**Soundness:** 3 good
**Presentation:** 3 good
**Contribution:** 3 good
**Rating:** 6
**Confidence:** 3

**Summary:**

This paper introduces the "MIntRec2.0" dataset, which offers a significant advancement in the field of multimodal intent recognition, addressing critical gaps in existing benchmarks by including multi-turn conversations, out-of-scope (OOS) utterances, and multi-party interactions. The dataset, derived from three TV series, encompasses 12.3 hours of dialogue across 1,245 conversations, totaling 15,000 annotated utterances. The expanded intent classes and inclusion of OOS labels mark a notable progression from previous datasets, aiming to bridge the gap between current benchmarks and real-world conversational scenarios.

**Strengths:**

- Dataset: The dataset's scale and inclusion of multi-party dialogues with both in-scope and OOS samples enhance its realism and applicability in human-computer interaction research.
- Approach: The framework efficiently handles multimodal data, and the detailed annotation, including speaker information, enriches its utility for diverse conversational research.
- Experiment: The comparison of human performance with state-of-the-art multimodal approaches in the dataset highlights the existing gap and provides a challenging benchmark for future research.

**Weaknesses:**

- The dataset's sourcing from only three TV series restricts the diversity of scenes and topics. This limitation might not fully represent the vast array of real-world conversational contexts.
- The unclear performance improvement in some metrics for multimodal fusion needs a more explicit explanation. This clarification will support the dataset's effectiveness in demonstrating the advantages of multimodal approaches.
- Additional experiments could better demonstrate the dataset's effectiveness, particularly in the nuances of multimodal intent recognition.
- The lack of detailed discussion on the incorporation and impact of multi-modal information, especially regarding out-of-scope data, is a notable omission. More detailed analyses or case studies would illuminate the challenges and benefits of using multi-modal data.
- The comparison between ChatGPT and human evaluations could benefit from more detail, such as the consistency of dialogue samples across experiments and how results might vary across different intent classes.

**Questions:**

1. Why is the improvement from using non-verbal multimodality not more pronounced? What are the key challenges in leveraging these modalities to enhance intent recognition accuracy?
2. Considering the complexity of intent in human interactions, what are the major influencing factors that the dataset and framework account for, and how are ambiguous or contradictory multimodal signals handled?
3. What are the significant differences between datasets focused on human-computer interactions and the proposed human-human interaction dataset in terms of multimodal intent classification tasks?
4. How do factors like laugh tracks or audience reactions in the TV series-based dataset influence the intent recognition process, and how do these sources differ from real-world multi-turn conversations?
5. Were annotators asked about which modalities were most valuable in understanding conversational intent? Such insights could help compare human perception with multimodal intent recognition systems.

---

> ### Author Response · Authors · 2023-11-18
> **Response to Reviewer Un9C (Part 1/7)**
>
> Thanks for your suggestions. We will address your concerns point by point.
>
> **Reply to Weaknesses**:
>
> **A1**. Thank you for your valuable feedback regarding the diversity of scenes and topics in our dataset. The first multimodal intent recognition dataset, MIntRec [1], is based on the TV series Superstore, featuring a rich array of characters and a multitude of storylines set in various scenes like shopping malls, warehouses, and offices. This dataset also presents hierarchical intent taxonomies with two broad categories, *Express Emotions or Attitudes* and *Achieve Goals*, which are grounded in philosophical theories [2], and delineates 20 fine-grained classes. Nonetheless, its data scale is somewhat limited.
>
> To address this and provide a more comprehensive resource for multimodal intent research, we introduced MIntRec2.0. This enhanced version incorporates two additional popular TV series, Friends and The Big Bang Theory. With 34 main characters and more than 10 primary types of conversational scenes and 15 distinct topics, MIntRec2.0 significantly enriches the dataset in terms of content. It covers a wide range of common intents encountered in daily life and expands the existing 20 classes to 30. Specifically, these conversational scenes and topics are diverse and include:
>
> **a. Scenes and Settings**:
> *Superstore* provides a unique retail environment with scenes in the store, cash registers, break room, parking lot, managerial offices, and warehouse, reflecting workplace dynamics and customer interactions.
> *Friends* showcases diverse social settings like the Central Perk, apartments, travel locations, and various city spots, emphasizing personal and relational interactions.
> *The Big Bang Theory* offers academic and living spaces, including apartments and the university, highlighting intellectual and social engagements.
>
> Each of these series brings a unique set of environments and interaction dynamics, ranging from personal and intimate to professional and public. The diversity in character backgrounds, professions, and social settings across these shows ensures a wide-ranging exploration of human interactions and conversational intents.
>
> Moreover, these series are culturally iconic and have significantly influenced societal communication patterns, making them highly relevant for studying contemporary conversational trends. Their popularity also ensures that the dataset is relatable and accessible for a broad range of researchers and applications.
>
> **b. Topics and Themes**:
> *Superstore* touches on workplace relations, management challenges, customer service scenarios, labor issues, and social issues like immigration and corporate dynamics.
> *Friends* explores friendship, romantic relationships, career challenges, and urban living, offering insights into a variety of emotional and relational topics.
> *The Big Bang Theory* delves into scientific discourse, geek culture, technological advancements, scientific research, social awkwardness, and the balance between intellectual pursuits and everyday life.
>
> The combination of these series presents an extensive range of human experiences and topics, from the mundane to the complex. This diversity enriches our dataset, making it an invaluable tool for studying and understanding the nuances of multimodal intent recognition in varied conversational contexts.
>
> Furthermore, these series, with their wide cultural impact, provide a relatable and realistic reflection of contemporary social dynamics, essential for developing robust and applicable AI models in the field of human-computer interaction.
>
> In summary, the chosen TV series offer a balanced mix of scenes and topics, providing a comprehensive resource that captures the complexity of human interactions and conversational intents. We are confident that our dataset's scope and diversity significantly contribute to the advancement of multimodal intent recognition research.
>
> **A2 \& A3**. Thank you for your insightful feedback. In our experiments, incorporating nonverbal modalities with multimodal fusion methods resulted in noticeable performance improvements. Specifically, we observed improvements of 1-4% in in-scope classification and 1-8% in both in-scope and out-of-scope utterance identification in single-turn conversations. To further demonstrate the effectiveness of nonverbal modalities, we conducted experiments across 30 specific intent classes and one out-of-scope category, using the F1-score as the evaluation metric, similar to [1]. The average performance over 5 experimental runs is presented below:

---

> ### Author Response · Authors · 2023-11-18
> **Response to Reviewer Un9C (Part 2/7)**
>
> **a. Single-turn Conversations**:
> |Methods| Acknowledge | Advise | Agree | Apologise | Arrange | Ask for help | Ask for opinions | Care | Comfort | Complain |
> | --- | --- | --- | --- | --- | --- | --- | --- | --- | --- | --- |
> |Text| 49.51 | 55.03 | 50.45 | 88.96 | 48.34 | 57.97 | 47.89 | 45.98 | 43.55 | 31.02 |
> |MAG-BERT| **54.86** | 53.86 | **54.54** | **93.02** | **52.04** | **61.14** | **48.89** | **50.05** | 41.69 | **32.26** |
>
> |Methods| Confirm | Criticize | Doubt | Emphasize | Explain | Flaunt | Greet | Inform | Introduce | Invite |
> | --- | --- | --- | --- | --- | --- | --- | --- | --- | --- | --- |
> |Text| 46.38 | 33.84 | 52.38 | 1.10 | 46.43 | 13.41 | 74.59 | 40.83 | 30.26 | 27.85|
> |MAG-BERT| 45.68 | **37.24** | 50.68 | **1.18** | **47.80** | **23.70** | **77.09** | **41.25** | **31.80** | **47.36** |
>
> |Methods| Joke | Leave | Oppose | Plan | Praise | Prevent | Refuse | Taunt | Thank | Warn | OOS |
> | --- | --- | --- | --- | --- | --- | --- | --- | --- | --- | --- | --- |
> |Text | 3.14 | 47.26 | 56.88 | 51.43 | 61.03 | 52.41 | 20.64 | 12.61 | 91.12 | 18.97 | 30.40 |
> |MAG-BERT| **8.85** | **47.61** | 56.60 | **54.73** | **63.30** | 51.32 | **29.37** | **13.76** | **91.85** | **30.89** | **34.03** |
>
> Our results show significant enhancements in understanding **26** intent classes when integrating nonverbal modalities. Notably, **14 classes saw improvements of over 3 points**, including *Acknowledge, Agree, Apologise, Arrange, Ask for help, Care, Criticize, Flaunt, Invite, Joke, Plan, Refuse, Warn*, and *OOS*. These classes represent a mix of common and challenging scenarios, as well as out-of-scope instances, all requiring high-level cognitive inference and semantic understanding. Remarkably, we observed substantial improvements of over **10** points in challenging classes like *Flaunt* ($\uparrow$ 10.29), *Invite* ($\uparrow$ 19.51), and *Warn* ($\uparrow$ 11.92). This underscores the importance of nonverbal modalities in recognizing human intentions. While there are also classes that show lesser improvements or rely more on text modality, the performance with non-verbal modalities is competitive, demonstrating the substantial benefit they bring to intent recognition in multimodal scenarios.
>
> Moreover, we extended our research to multi-turn conversations, maintaining the same experimental settings as in single-turn conversations.
>
> **b. Multi-turn Conversations**:
>
> |Methods| Acknowledge | Advise | Agree | Apologise | Arrange | Ask for help | Ask for opinions | Care | Comfort | Complain |
> | --- | --- | --- | --- | --- | --- | --- | --- | --- | --- | --- |
> |Text| 55.04 | 55.28 | 51.43 | 91.65 | 48.00 | 56.94 | 45.71 | 55.76 | 48.98 | 40.30 |
> |MAG-BERT| **61.67** | **55.45** | **53.22** | **91.98** | **52.78** | **60.65** | **51.88** | **56.72** | **50.50** | **40.82** |
>
> |Methods| Confirm | Criticize | Doubt | Emphasize | Explain | Flaunt | Greet | Inform | Introduce | Invite |
> | --- | --- | --- | --- | --- | --- | --- | --- | --- | --- | --- |
> |Text| 47.66 | 36.32 | 49.89 | 2.86 | 47.28 | 15.46 | 78.15 | 43.40 | 32.85 | 39.58 |
> |MAG-BERT| 46.85 | **37.22** | **51.46** | 2.11 | **48.60** | 8.34 | **79.43** | **45.64** | 32.35 | 36.46 |
>
> |Methods| Joke | Leave | Oppose | Plan | Praise | Prevent | Refuse | Taunt | Thank | Warn | OOS |
> | --- | --- | --- | --- | --- | --- | --- | --- | --- | --- | --- | --- |
> |Text| 4.98 | 53.43 | 58.00 | 51.91 | 66.23 | 56.90 | 31.60 | 15.33 | 84.72 | 27.01 | 63.56 |
> |MAG-BERT| 4.66 | 52.01 | 55.87 | 48.44 | **67.65** | 46.67 | 21.36 | 13.27 | **85.23** | **28.19** | 62.52 |
>
> In these settings, MAG-BERT outperformed text-only modality in 18 classes. Specifically, it achieved improvements of over 3 points in four classes, including *Acknowledge, Arrange, Ask for help*, and *Ask for opinions*, and 1-2% improvements in 8 classes, such as *Agree, Comfort, Doubt, Explain, Greet, Inform, Praise*, and *Warn*. These classes encompass a significant portion of common interaction intents. However, the gains from nonverbal modalities in multi-turn conversations were not as pronounced as in single-turn conversations, indicating existing methodological limitations in handling out-of-scope utterances and fully utilizing context information. Addressing these challenges is crucial for future research and highlights both the importance and complexity of the MIntRec2.0 dataset. We would like to include these experimental results and discussions in the appendix of the revised paper.
>
> **A4**. Thank you for your insightful question. To address the impact of incorporating multimodal information in intent recognition, we first carried out additional experiments to examine the fine-grained performance of each class using both text-only and multimodal fusion baselines (as detailed in responses to **Reply to Weaknesses**-A2\&A3).

---

> ### Author Response · Authors · 2023-11-18
> **Response to Reviewer Un9C (Part 3/7)**
>
> Second,  to further illustrate the impact of multimodal information in intent recognition, we selected a specific dialogue from our dataset for a detailed case study. This allowed us to compare the predictive accuracy of both text and MAG-BERT models for each utterance, considering speaker identity and ground truth. The results are as follows:
>
> |Index|Speaker|Utterance|Predicted Label / Confidence (TEXT)|Predicted Label / Confidence (MAG-BERT)| Ground Truth |
> | --- | --- | --- | --- | --- | --- |
> |0| Glenn |salvatore kazlauskas.|**Greet** / 0.0735| OOS / 0.8289 | OOS |
> |1| Nico | wait, you mean creepy sal? |Confirm / 0.9563| Confirm / 0.6321 | Confirm |
> |2| Glenn | the man is dead. |Inform / 0.4575| Inform / 0.3950 | Inform |
> |3| Dina | police said he's been dead for at least a year. |Inform / 0.9842| Inform / 0.6941 | Inform |
> |4| Amy | are you crying? |**Care** / 0.7637| **Doubt** / 0.3580 | Confirm |
> |5| Kelly | poor guy. |**Taunt** / 0.1573| **Taunt** / 0.1045 | OOS |
> |6| Amy | but you didn't know him. |**Comfort** / 0.0932| **Explain** / 0.1131 | OOS |
> |7| Kelly | but he was a human being. |**Emphasize** / 0.0693| **Introduce** / 0.0603 | OOS |
> |8| Cheyenne | when he looked at you, it felt like he was grabbing you. |**Complain** / 0.3858| **Complain** / 0.1209 | OOS |
> |9|Glenn | apparently he was doing some work behind the drywall outside the women's washroom and then his foot got caught in a beam, and he starved to death. |Inform / 0.3145| Inform / 0.5194 | Inform |
> |10|Glenn| we're not sure. |**Inform** / 0.1238| **Inform** / 0.1233| OOS |
> |11|Glenn| he drilled a hole into the women's washroom so... |Inform / 0.2779| Inform / 0.1861|Inform|
> |12|Glenn| why? |Doubt / 0.9722| Doubt / 0.8096 | Doubt |
> |13|Dina| i know we all assumed that was amy. |**Emphasize** / 0.0682| Explain / 0.0996 | Explain |
> |14|Amy| why--why me? |Doubt / 0.7256| Doubt / 0.6292 | Doubt |
> |15|Dina| cause, you know... |Explain / 0.9794| Explain / 0.4273 | Explain |
> |16|Dina| i'm sure there's a lot of churn going on in there. |**Comfort** / 0.1254| **Explain** / 0.0841 | Taunt |
> |17|Cheyenne| wait, so it's just going to sit in the store? |Doubt / 0.9335| Doubt / 0.8337 | Doubt |
> |18|Nico| uh...   i'm not working next to a dead body. |**Oppose** / 0.9767| **Oppose** / 0.3158 | Taunt|
> |19|Dina| technically we've all been working next sal's dead body for the past year. |**Explain** / 0.1855| **Explain** / 0.22875768 | Inform |
> |20|Dina| nobody complained until now. |**Emphasize** / 0.0534| **Inform** / 0.0770| OOS |
> |21|Jonah| that must've been sal's foot we found. |**Inform** / 0.078| **Introduce** / 0.1448 | OOS |
> |22|Dina| actually he still had both of his feet. |**Doubt** / 0.0581| **Inform** / 0.0656 | Oppose |
>
>
> The data reveals that MAG-BERT generally achieves high accuracy in most in-scope classes, except for some with complex semantics like Taunt, Oppose, and Confirm. Furthermore, in many correctly predicted in-scope classes, MAG-BERT demonstrates a high confidence level, often exceeding a 0.5 probability. In contrast, the TEXT model showed more errors than MAG-BERT in two utterances and tended to exhibit higher confidence in certain utterances (e.g., index 4, 5, 8, 18). This comparison highlights the effectiveness of incorporating non-verbal modalities over relying solely on textual information. However, it's important to note that MAG-BERT struggles with out-of-scope (OOS) utterances, often making errors in these categories. This observation suggests that while existing multimodal fusion methods have capabilities in recognizing known intents, their performance in detecting out-of-scope utterances is limited, pointing to a significant area for future research and development.
>
> Finally, we conducted an analysis of the composition of out-of-scope utterances. Upon examining a random sample of 100 OOS utterances, we identified several utterances that potentially belong to new intent classes with limited examples, such as *help*, *drive person away*, *guess*, and *wish*. We also noted that a significant portion of the sample (nearly 60%) consists of utterances that can be categorized as *statement-opinion*, where the speaker expresses personal views on a subject or individual. Another notable segment (nearly 20%) falls under *statement-non-opinion*, encompassing subjective statements. These findings correlate with the 42 dialogue act classes defined in the SwDA dataset [3] and align with our discussion in the main paper's Section 3, particularly in the subsection on out-of-scope utterances. Our study indicates the need for continuous exploration in this field to effectively identify and categorize a broader range of human intents, especially those that fall outside the currently defined scope.

---

> ### Author Response · Authors · 2023-11-18
> **Response to Reviewer Un9C (Part 4/7)**
>
> **A5**. Thank you for your insightful question. In our experiments comparing ChatGPT and humans, the set of 10 dialogues totaling 227 utterances remained consistent across all experiments to ensure a fair comparison between ChatGPT and human participants. The performance metrics for each method were aggregated across all intent classes. To delve deeper into the performance on specific intent classes, we conducted additional experiments using the same datasets for ChatGPT-10, Humans-10, and Humans-100 as outlined in Table 5. We calculated the F1-score for each intent class and have presented these results as follows:
>
> |Methods| Acknowledge | Advise | Agree | Apologise | Arrange | Ask for help | Ask for opinions | Care | Comfort | Complain |
> | --- | --- | --- | --- | --- | --- | --- | --- | --- | --- | --- |
> |ChatGPT-10| 24.44 | 30.84 | 35.29 | 62.94 | 17.39 | 28.39 | 22.95 | 0.00 | 40.00 | 34.16 |
> |Humans-10|46.27 | 67.76 | 61.90 | 93.02 | 55.46 | 69.33 | 50.00 | 50.57 | 66.67 | 49.54 |
> |Humans-100| **69.23** | **70.73** | **65.67** | **93.02** | **64.22** | **79.45** | **61.11** | **62.34** | **77.23** | **64.11** |
>
> |Methods| Confirm | Criticize | Doubt | Emphasize | Explain | Flaunt | Greet | Inform | Introduce | Invite |
> | --- | --- | --- | --- | --- | --- | --- | --- | --- | --- | --- |
> |ChatGPT-10|25.00 | 14.12 | 15.05 | 6.45 | 33.55 | 18.46 | 64.52 | 31.76 | 8.22 | 47.06 |
> |Humans-10|60.19 | 59.05 | 60.67 | 31.03 | 55.67 | 40.00 | 86.13 | 50.41 | 47.76 | 38.71 |
> |Humans-100|**62.43** | **67.96** | **68.35** | **42.86** | **65.79** | **67.92** | **90.91** | **67.09** | **66.67** | **70.97** |
>
> |Methods| Joke | Leave | Oppose | Plan | Praise | Prevent | Refuse | Taunt | Thank | Warn | OOS |
> | --- | --- | --- | --- | --- | --- | --- | --- | --- | --- | --- | --- |
> |ChatGPT-10|16.00 | 33.33 | 25.00 | 29.27 | 50.00 | 12.12 | 16.87 | 13.48 | 59.09 | 37.04 | 27.85 |
> |Humans-10|28.17 | 71.30 | 67.51 | 57.78 | 73.06 | 70.77 | 47.83 | 44.62 | 93.91 | 34.78 | 62.83 |
> |Humans-100| **50.00** | **79.28** | **67.56** | **70.89** | **79.65** | **77.97** | **53.66** | **63.93** | **94.83** | **62.86** | **75.41** |
>
> The results illustrate that humans significantly outperform ChatGPT. With the same foundational knowledge of 10 dialogues encompassing 227 utterances, Humans-10 exhibited superior performance across nearly all intent classes and the out-of-scope category, outperforming ChatGPT-10 by over 10 points in most cases. **Notably, 15 intent classes and one out-of-scope category saw improvements of over 30 points. Classes like *Care* and *Prevent* achieved improvements of over 50 points, and *Ask for help*, *Criticize*, *Doubt*, and *Oppose* saw over 40 points improvement.** These findings highlight a significant gap between ChatGPT and human capabilities, underscoring humans' adeptness at using limited prior knowledge from multimodal contexts, such as body language and facial expressions, to infer and synthesize complex intents at a cognitive level—a skill where current machine learning methods, including large language models, fall short.
>
> Additionally, with more extensive prior knowledge of 100 dialogues comprising 997 utterances, Humans-100 performed even better compared to Humans-10 and achieved state-of-the-art performance across all classes. **This included markedly improved performance of over 10 points in 16 intent classes and the out-of-scope category**. This demonstrates the remarkable potential of humans to leverage multimodal knowledge and their ability to learn effectively with only a marginally larger dataset (7% of all training data). This proficiency even surpasses current fully supervised multimodal fusion methods, as shown in Tables 4 and 5. The detailed intent performance comparison between ChatGPT and humans further validates the challenges presented by the MIntRec2.0 dataset, indicating that there is still considerable progress to be made in AI for the complex task of multimodal intent recognition. We plan to include these results in the Appendix of the revised paper.
>
> **Reply to Questions**:
>
> **A1**. Firstly, intent recognition has its roots in natural language processing, where numerous foundational works have focused on understanding the semantics of utterances, particularly within goal-oriented dialogue systems [4, 5, 6]. Consequently, it's unsurprising that text modality often serves as the primary means for interpreting multimodal language. This predominance of text is further evidenced in other well-known multimodal tasks like sentiment analysis and emotion recognition [7, 8, 9, 10, 11]. However, in real-world scenarios, the integration of non-verbal modalities becomes essential. Our world is inherently multimodal, and non-verbal cues significantly enhance the comprehension of true intentions, especially in open-ended dialogues. Recognizing the need for a high-quality, large-scale multimodal intent dataset, we developed MIntRec2.0 to bridge this gap and facilitate further research.

---

> ### Author Response · Authors · 2023-11-18
> **Response to Reviewer Un9C (Part 5/7)**
>
> Secondly, the incorporation of non-verbal modalities indeed yields notable improvements. As indicated in Table 4 of our main paper, multimodal fusion methods such as MAG-BERT and MulT demonstrate score enhancements of 1-4% across various metrics in both in-scope classification and out-of-scope detection. While these modalities might not always yield substantial improvements in certain aspects like handling out-of-scope utterances or contextual information, this is largely due to the absence of multimodal fusion methods specifically designed for these purposes. However, as detailed in **Reply to Weaknesses**-A2\&A3, MAG-BERT notably enhances performance across many intent classes, highlighting the significant potential of non-verbal modalities.
>
> Finally, the MIntRec2.0 dataset presents the first large-scale resource in multimodal intent research, offering a comprehensive framework for benchmarking with advanced multimodal fusion methods. Our extensive evaluations reveal that while non-verbal modalities contribute to recognizing complex cognitive-level intents, they face challenges in effectively leveraging context and out-of-scope utterances. This limits their robustness and generalizability in this demanding task. Comparing the performances of humans and ChatGPT, we observe that humans excel at using limited multimodal language as prior knowledge for synthesizing and inferring real intentions, significantly outperforming even sophisticated large language models. This distinction underscores the importance and challenges posed by this dataset, marking it as a valuable contribution to the research community.
>
> **A2**. In addressing the complexity of understanding a speaker's intentions during interactions, two main aspects emerge: non-verbal information and background information. The first aspect involves effectively leveraging non-verbal modalities, such as video and audio, to capture cross-modal interactions. The second aspect encompasses various types of side information, including conversational context, out-of-scope utterances, and other objective factors (e.g., cultural and social background, scenes, and topics). These elements serve as implicit prior knowledge, aiding in the understanding of human intentions. A detailed introduction of these perspectives is as follows:
>
> * **Non-Verbal Cues**: Non-verbal modalities such as video and audio are also significant for understanding human intentions in multimodal scenarios. Tones, gestures, facial expressions, body language, and eye contact are crucial in providing additional information about the intention of a speaker. These cues can add to, contradict, or enhance the verbal information, offering deeper insights into the speaker's real intentions.
> * **Background Information**
>     - **Conversational Context**: This refers to the specific flow of information within the dialogue and the relevance of previous utterances to the current one. Understanding the conversational history is crucial for interpreting the speaker's current intentions. The context may include the topics being discussed, the progress of the dialogue interactions, and the shared knowledge and experiences between the speakers, which are significant in helping understand intentions.
>     - **Out-of-scope Utterances**: The occurrence of statements or new intentions that deviate from the known intentions of the speaker can help discover new intents. On the one hand, understanding out-of-scope utterances helps avoid misclassifying them as in-scope intents. On the other hand, detected out-of-scope utterances (e.g., unrelated or new topics) can be used to understand broader communicative goals, facilitate human communication, and identify potential communication interests.
>     - **Objective Factors**: (1) Cultural and Social Background: The cultural norms, social backgrounds, and linguistic styles of the speakers influence how intentions are expressed and perceived. This includes factors like accent, dialect, and the social context of the conversation. (2) Scenes and Topics: The physical setting of the conversation and the subject matter can have a significant impact on a speaker's intent. Different environments and topics can elicit varied communicative intents and styles.

---

> ### Author Response · Authors · 2023-11-18
> **Response to Reviewer Un9C (Part 6/7)**
>
> Regarding ambiguous or contradictory multimodal signals, we prioritize the clarity of video content, ensuring it contains clear textual utterances and discernible facial or body language. Contradictory signals across different modalities highlight the complexity of understanding human intent, necessitating nuanced analysis and inference of multimodal synergy. We preserve these multifaceted signals in their natural form to maintain the dataset's real-world applicability and generalizability. Hence, MIntRec2.0 represents a significant step forward in researching human intention understanding, especially in tackling the inherent challenges posed by ambiguous and complex multimodal interactions.
>
> **A3**. The difference between human-computer and human-human interaction datasets is substantial, primarily due to the inherent differences in the nature of communication and interaction in each setting. Here are the primary three distinctions:
> * **Interaction Dynamics**: In human-computer interactions, the dynamics are typically unidirectional or asymmetrical. Users often lead the conversational directions and give commands to generate dialogue utterances (e.g., in goal-oriented dialogue systems [4, 5]). In contrast, human-human interactions are more dynamic and bidirectional (e.g., in open-ended dialogue systems [12]), with both parties actively contributing, responding, and adapting to the conversation flow.
> * **Complexity of Communication**: Human-computer interactions are generally more structured and predictable, with a limited range of intents that follow specific orders or needs and relatively simple responses. Human-human interactions are far more complex, involving a wider range of intents, subtleties, emotions, and unpredictability.
> * **Non-Verbal Cues**: Non-verbal cues are often limited or absent in human-computer interactions, as seen in many task-oriented datasets in NLP [4, 5, 6]. Even with advanced multimodal fusion algorithms, interpreting non-verbal cues from humans remains challenging for computers. In human-human interactions, non-verbal cues play a crucial role in understanding intent, with nuances in body language, facial expressions, and tone carrying significant information.
>
> Therefore, constructing a human-human interaction dataset for multimodal intent recognition is more appropriate, as it satisfies interaction dynamics, supports complex communication, and incorporates non-verbal cues. MIntRec2.0 makes a pioneering contribution in this area and aims to facilitate related research and application.
>
> **A4**. Thank you for your question. In the process of data collection and dialogue segmentation from the TV series, we indeed encountered laugh tracks and other audience signals following the utterances. To ensure data integrity and focus on the complexity of the intents, we meticulously annotated only the speaking parts, excluding these external audience cues. This approach helps maintain data quality by providing relatively pure multimodal signals for analysis.
>
> Regarding the TV series as a data source, they are derived from real-world, open-ended, multi-turn conversational scenarios with a diverse range of scenes and topics (as detailed in our response to Reviewer SFq3-Reply to Weaknesses-A1). This makes them substantially similar to multi-turn conversations in real-world contexts, offering a rich resource for gathering human intents from everyday interactions.
> However, a primary distinction lies in the scope of intent categories defined in our study. The 30 intent categories we identified are based on existing hierarchical multimodal intent taxonomies [1], encompassing 16 classes in *Express Emotions or Attitudes* and 14 classes in *Achieve Goals*. These categories, which are prevalent in daily life, were determined after extensive analysis and summarization of dialogue videos. They theoretically align with philosophical bases of intent, encompassing both the traditional artificial intelligence definitions of intent (as plans or goals of an agent, coupled with corresponding feedback actions [13, 14]) and the emotional evaluation factors influenced by the brain [2]. However, we recognize that some real-world intent classes may not be fully covered by our defined categories. To address this, we included an additional out-of-scope (OOS) category for utterances that do not fit into any known classes. This category facilitates the exploration of new potential intents and, with further real-world data, can aid in broadening the scope of intent classification beyond a closed-world framework.
>
> In summary, our intent detection system is designed to be adaptable to real-world scenarios, and the chosen data sources provide a solid foundation for future research into human intent recognition.

---

> ### Author Response · Authors · 2023-11-18
> **Response to Reviewer Un9C (Part 7/7)**
>
> **A5**. Thank you for raising this important question. Firstly, it's crucial to note that intent recognition primarily stems from natural language processing, with significant foundational research and advancements in this area [4, 5, 6], including high performance in some goal-oriented dialogue systems. As a result, the text modality often plays a central role in deciphering complex human intentions, as evidenced in [1]. Nevertheless, in real-world settings, the integration of non-verbal modalities, such as video and audio, along with conversational context and scene information, is crucial to accurately infer intentions. Recognizing the limitations of existing datasets, which are often small-scale, single-turn, and limited to closed-world classification, our MIntRec2.0 dataset aims to facilitate research into the effectiveness of non-verbal modalities in multi-turn conversations and in contexts that more closely resemble real-world situations, including out-of-scope utterances.
>
> In the process of multimodal intent annotation, our annotators utilized textual, video, and audio information, combined with contextual data within each dialogue, to analyze the underlying intention of each utterance. To investigate the relative importance of different modalities in understanding conversational intent, we gathered insights on key aspects including text modality (*spoken language*), video modality (*facial expressions* and *body language*), audio modality (*tone*), and background (*context information* and *conversational scenes*). We asked 9 annotators involved in both annotation and human evaluation to rank these six aspects based on their importance in understanding human intentions, drawing from their experience. The aggregated ranking results are as follows:
>
> |Rank|Spoken Language| Facial Expressions | Tone | Context Information | Body Language | Conversational Scenes |
> | --- | --- | --- | --- | --- | --- | --- |
> |1| **6** | 0 | 2 | 1 | 0 | 0 |
> |2| 2 | **4** | 2 | 1 | 0 | 0 |
> |3| 1 | 3 | **3** | 1 | 1 | 0 |
> |4| 0 | 2 | 1 | **4** | 2 | 0 |
> |5| 0 | 0 | 0 | 2 |  **6**| 1 |
> |6| 0 | 0 | 0 | 0 | 1 | **8** |
>
> These results suggest that *spoken language* is the most critical factor, followed by *facial expressions, tone, context information, body language*, and *conversational scenes*. While *spoken language* is predominant, non-verbal cues like *facial expressions* and *tone* are valuable in perceiving emotions or attitudes, especially in the *Express Emotions or Attitudes* category. *Context information* provides essential background knowledge about the speakers, aiding in a more profound understanding of intentions. *Body language*, though complex and implicit, is also insightful, particularly in the *Achieve Goals* category. Conversational scenes, while less critical in open-ended dialogues, still contribute to understanding intent in specific contexts. We plan to include these insights and discussions in the Appendix of the revised paper to provide a more comprehensive understanding of the multimodal intent recognition process.
>
> **References**:
>
> [1] MIntRec: A New Dataset for Multimodal Intent Recognition. Zhang et al. ACM MM 2022.
> [2] Intention, emotion, and action: A neural theory based on semantic pointers. Schröder et al. Cognitive science. 2014.
> [3] Switchboard: Telephone speech corpus for research and development. Godfrey et al. ICASSP. 1992.
> [4] Snips Voice Platform: An Embedded Spoken Language Understanding System for Private-by-design Voice interfaces. Coucke et al. arXiv. 2018.
> [5] An evaluation dataset for intent classification and out-of-scope prediction. Larson et al. EMNLP-IJCNLP 2019.
> [6] Efficient Intent Detection with Dual Sentence Encoders. Casanueva et al. arXiv. 2020.
> [7] Learning Modality-Specific Representations with Self-Supervised Multi-Task Learning for Multimodal Sentiment Analysis. AAAI 2021.
> [8] Improving Multimodal Fusion with Hierarchical Mutual Information Maximization for Multimodal Sentiment Analysis. Han et al. EMNLP 2021.
> [9] MOSI: Multimodal Corpus of Sentiment Intensity and Subjectivity Analysis in Online Opinion Videos. Zadeh et al. arXiv. 2016.
> [10] Multimodal Language Analysis in the Wild: CMU-MOSEI Dataset and Interpretable Dynamic Fusion Graph. Zadeh et al. ACL 2018.
> [11] MELD: A Multimodal Multi-Party Dataset for Emotion Recognition in Conversations. Poria et al. ACL 2019.
> [12] IEMOCAP: interactive emotional dyadic motion capture database. Busso et al. Language resources and evaluation. 2008.
> [13] Intention,–Plans,–and–Practical–Reason. Michael E Bratman. Mind. 1988.
> [14] Intelligent agents: Theory and practice. Michael Wooldridge and Nicholas R Jennings. The knowledge engineering review. 1995.

---

### Author Response · Authors · 2023-11-20
**Author Response Clarifications (1 / 2)**

# Summary of Our Response and Revision

We express our gratitude to all reviewers for their valuable time and constructive feedback. All concerns have been meticulously addressed, with revisions highlighted in blue in the updated version.
We are heartened by the positive evaluations of our work, "extensive scale, diversity of dialogues, and thorough annotation, providing a realistic and challenging benchmark for human-computer interaction research" (**Un9C**), "closeness to real-world scenarios" (**SFq3**), and "interesting and valuable contribution to human-machine conversational interactions, clearly motivated" (**kGfg**), including being "the first to incorporate multi-party dialogues and OOS labels" (**ju3D**).

## 1. General Clarifications
* Contributions
    - The dataset includes over 10 coarse-grained scenes and 15 topics from three TV series, totaling 12.3 hours, and comprises 1,245 dialogues and 15,040 utterances from real-world scenarios.
    - Detailed annotations are provided for 30 specific intents and one out-of-scope category, complete with precise timestamps, speaker identities for 34 main characters, and well-defined dialogue divisions.
    - A versatile framework has been developed, supporting both single-turn and multi-turn conversations, encompassing in-scope classification and out-of-scope detection.
    - Comprehensive evaluations are conducted on text, multimodal fusion methods, ChatGPT, and human performances. These evaluations reveal the effectiveness of non-verbal modalities but also highlight a substantial gap in high-level cognitive understanding between existing AI methods and human capabilities, particularly in complex intent recognition.
    - MIntRec2.0 is the first extensive multimodal, multi-party intent resource tailored for real-world human-machine conversational interactions. This pioneering dataset is expected to significantly advance related research and applications.

To thoroughly assess the proposed benchmark dataset, we explore two key aspects:

* Text and Multimodal Fusion Method Evaluations:
    - Single-Turn Conversations: Initial training uses only in-scope utterances, followed by testing with both in-scope and out-of-scope utterances. Multimodal methods like MAG-BERT and MulT show improvements of 1-4% in scores. Fine-grained analysis reveals enhancements in 26 classes, with 14 classes showing over 3 point improvements. However, inclusion of out-of-scope utterances yields marginal improvements.
    - Multi-Turn Conversations: Training incorporates context information, including out-of-scope utterances. Multimodal methods exhibit lesser improvements over text in comparison to single-turn conversations. Fine-grained analysis indicates improvements in 18 classes, with 8 classes showing over 1% enhancements. A case study reveals that while multimodal methods accurately recognize most in-scope classes, they frequently err in categorizing out-of-scope utterances. These findings suggest significant challenges for current methods in leveraging context and out-of-scope data for enhanced multimodal representation learning.
* ChatGPT and Human Baseline Evaluations:
    - ChatGPT: Two settings were explored: one with no prior knowledge (ChatGPT-0) and the other with limited prior knowledge of 10 dialogues (ChatGPT-10). While ChatGPT demonstrates strong generalization abilities in a zero-shot setting, it struggles with limited training data.
    - Humans: Ten participants conducted human evaluations on the test set. Initially provided with the same prior knowledge as ChatGPT (Humans-10), they predict intents on a subset of the test data. Results indicate that humans significantly outperform ChatGPT across all metrics, with over 30 score improvements. Fine-grained analysis shows over 30 score improvements in 16 classes, with nearly all classes showing over 10 score improvements. With additional knowledge of 100 dialogues (7% of the total training data), humans display 7-13% improvements over Humans-10 and achieve the state-of-the-art performance of 71\% accuracy.

These results underscore the efficacy of non-verbal modalities in enhancing intent recognition but also reveal shortcomings in addressing out-of-scope utterances or contextual information, impacting model robustness. The substantial performance gap between AI methods and humans underscores the significance and challenges of the proposed multimodal intent recognition task and benchmark dataset.

---

> ### Author Response · Authors · 2023-11-20
> **Author Response Clarifications (2 / 2)**
>
> ## 2. New Results in Revision
> 1. We fine-grained intent performance results in single-turn and multi-turn conversations, illustrating the effectiveness of non-verbal modalities (page 31, Appedix U.1, **Un9C**, **SFq3**, **ju3D**).
> 2. We conduct a case study to demonstrate the effectiveness of non-verbal modalities, especially in out-of-scope data (page 30, Appendix T, **Un9C**, **ju3D**).
> 3. We fine-grained intent performance of ChatGPT-10, Humans-10, Humans-100, highlighting the significant gap between humans and AI methods (page 31, Appendix U.2, **Un9C**, **ju3D**).
> 4. We add voting results on the significance of modalities during multimodal intent annotation (page 22, Appendix L, **Un9C**, **ju3D**).
>
> ## 3. Revisions to the Text
> 1. We introduce scenes and topics in the dataset to demonstrate data diversity (page 18, Appendix E, **Un9C**, **SFq3**).
> 2. We correct minor typos throughout the text (pages 2, 5, **SFq3**, **Un9C**).
> 3. We discuss the differences between human-computer and human-human interaction datasets  (page 16, Appendix C, **Un9C**, **SFq3**).
> 4. We clarify the composition of out-of-scope data (page 5, **Un9C**, **ju3D**).
> 5. We clarify there are no audience signals and the consistency of prior knowledge in the Humans-10 and ChatGPT-10 evaluations (pages 4, 8, **Un9C**, **ju3D**).
>
> **Each reviewer's concerns have been individually addressed. We humbly request careful review and feedback on our responses and revised work and sincerely appreciate your support during this review process.**

---

### Meta-Review · Area_Chair_nz25 · 2023-12-06

**Metareview:**

This paper presents a new multi-modal intent recognition dataset on multi-modal conversational interactions, formed of interactions from three TV series covering topics from daily life and corresponding subtitles from OpenSubtitles. A special focus is given in the collection to inclusion of out-of-domain/-scope samples. The work includes evaluation benchmarks with ChatGPT, classical multi-mofal fusion methods, as well as human evaluation. The paper is written clearly, and the rebuttal includes long and detailed responses addressing reviewer comments and suggestions,

**Justification For Why Not Higher Score:**

Tha data mainly includes acted interactions, and lack of real data is a limitation.

**Justification For Why Not Lower Score:**

Although it has limitations, the dataset and the proposed benchmark and framework will empower future research.

---

### Decision · Program_Chairs · 2024-01-16

Accept (poster)